



# Spatial distributions of iron and manganese in surface waters in the Arctic's Laptev and East Siberian seas

Naoya Kanna[1*], Kazutaka Tateyama[2], Takuji Waseda[3], Anna Timofeeva[4], Maria Papadimitraki[5, 6], Laura Whitmore[7], Hajime Obata[1], Daiki Nomura[8, 9, 10], Hiroshi Ogawa[1], Youhei Yamashita[11], Igor Polyakov[7]

[1]Atmosphere and Ocean Research Institute, The University of Tokyo, 5-1-5, Kashiwanoha, Kashiwa-shi, Chiba 277-8564 Japan
[2]Kitami Institute of Technology, Kitami Hokkaido Japan
[3]Department of Ocean Technology, Policy and Environment, Graduate School of Frontier Sciences, The University of Tokyo, Kashiwa Chiba Japan
[4]Arctic and Antarctic Research Institute
[5]National Institute of Aquatic Resources-Technical University of Denmark
[6]Department of Biology, University of Southern Denmark, 5230 Odense M, Denmark
[7]International Arctic Research Center, University of Alaska Fairbanks
[8]Arctic Research Center, Hokkaido University, Kita-21 Nishi-11, Kita-ku, Sapporo-shi, Hokkaido 001-0021, Japan
[9]Global Station for Arctic Research, Global Institution for Collaborative Research and Education, Hokkaido University, Kita-21 Nishi-11, Kita-ku, Sapporo-shi, Hokkaido 001-0021, Japan
[10]Field Science Center for Northern Biosphere, Hokkaido University, 3-1-1 Minato-cho, Hakodate-shi, Hokkaido 041-0821, Japan
[11]Faculty of Environmental Earth Science, Hokkaido University, Kita-10 Nishi-5, Kita-ku, Sapporo, Hokkaido 060-0810, Japan

*Correspondence to*: Naoya Kanna (nkanna@g.ecc.u-tokyo.ac.jp)

Atmosphere and Ocean Research Institute, The University of Tokyo

**Abstract.** The Arctic's Laptev and East Siberian Seas (LESS) is a region with high biogeochemical activity. Nutrient inputs associated with river runoff and shelf sediment-water exchange processes are vital for supporting primary production in the LESS. However, the dynamics of trace metals such as iron (Fe) and manganese (Mn), which are essential micronutrients for primary producers, remain unknown in the LESS. Here, we present data on Fe and Mn in surface waters in the late summer of 2021 and discuss the factors controlling their concentrations and distributions on the surface of the LESS continental margins. Property of surface waters in the East Siberian Sea and Chukchi Abyssal Plain differed significantly from the Makarov and Amundsen Basins. Nutrient-rich Pacific-sourced water exists in the East Siberian Shelf with a high dissolved Mn (dMn) concentration. Pacific-sourced water also receives a large sedimentary flux that releases dMn onto the continental shelf. Dissolved Fe (dFe) could have been released on the shelf as well; however, dFe remained low in the Pacific-sourced water. This is because dFe re-precipitated more rapidly than dMn because of the difference in removal kinetics. In contrast, relatively nutrient-poor Atlantic-sourced water exists in the Makarov and Amundsen Basins. A positive correlation between the fraction





of meteoric water (river water and precipitation), dFe, and humic-like colored dissolved organic matter (CDOM) in Atlantic-sourced water confirmed a common freshwater source for dFe and CDOM. Terrigenous organic ligands likely stabilized Fe in the dissolved phase, which was not the case for Mn. The fraction of sea ice meltwater was not correlated with dFe and dMn in any part of the sampling domain. Our results indicate that the major factors controlling these metal concentrations in the LESS
continental margins are river discharge and the input of shelf sediment.

## 1 Introduction

The Arctic's Laptev and East Siberian Seas (LESS) is strongly influenced by ongoing changes in the Arctic climate. The LESS has shown a drastic increase in net primary production in recent decades as evidenced by satellite records (Lewis et al., 2020) as the region has warmed (Rantanen et al., 2022) and seasonal ice coverage has decreased (Fox-Kemper et al., 2021; Sumata
et al., 2023). Since the middle of the 2010s, the increased penetration of Atlantic Water into the LESS continental margins has driven "Atlantification", which weakened oceanic stratification, enhanced upward fluxes of heat and nutrients due to the increased vertical mixing, and reduced sea ice coverage (Polyakov et al. 2017, 2020, 2023). Given the increasing trend in the discharge of Eurasian rivers (Feng et al., 2021), the export of terrestrial materials, including nutrients and trace metals, to the LESS might intensify, and potentially affect biological production and carbon deposition on the shelves. The fluxes of shelf-
derived materials from the LESS to the Central Arctic have likely increased over the past decade (Kipp et al., 2018). Moreover, Arctic sea ice often contains sediments entrained on the East Siberian Arctic Shelf (Eiken et al., 2000, 2005; Hölemann et al., 1999a; Krumpen et al., 2020; Waga et al., 2022; Wegner et al., 2017), such that the decreasing trend in seasonal ice coverage might affect material fluxes to the Central Arctic upon melting. Therefore, the LESS is a key region for understanding how climate change impacts biogeochemical cycling in the Arctic Ocean.

Iron (Fe) and manganese (Mn) are essential micronutrients for primary producers, and are relevant to important phytoplankton metabolic pathways (Morel and Price, 2003; Twining and Baines, 2013). Fe and Mn are supplied to surface waters from common sources, such as porewaters of shelf sediments (Cid et al., 2012; Jensen et al., 2020; Kondo et al., 2016), sea ice meltwater (Bolt et al., 2020; Evans and Nishioka, 2019; Hölemann et al., 1999a, 1999b), and river waters (Guieu et al., 1996; Pokrovsky et al., 2016; Savenko and Pokrovsky, 2019). When paired, Fe and Mn concentrations can often be used as indicators
of common-source fluxes (Jensen et al., 2020; Landing and Bruland, 1987). Ongoing changes in the LESS may intensify the supply of Fe and Mn to surface waters in the Central Arctic. Some Fe and Mn derived from these sources are carried in the Central Arctic by the Transpolar Drift, a major current that directly transports Eurasian shelf water, river waters, and sea ice from the LESS (Charette et al., 2020; Gerringa et al., 2021). A few studies indicated the importance of Lena River runoff and sea ice melt on the Fe and Mn distributions near the Laptev Sea (Hölemann et al., 1999, 2005; Klunder et al., 2012; Middag et
al., 2011). Yet, the LESS is still one of the least-studied areas of the Arctic Ocean in terms of trace metal dynamics. In particular, little data regarding trace metal is reported over the East Siberian Sea.





This study reports the spatial distributions of Fe and Mn in the surface waters of the LESS including the East Siberian Sea (ESS) and Chukchi Abyssal Plain (CAP), Makarov Basin (MB), and Amundsen Basin (AB). Observations were made in international cooperation with the Nansen and Amundsen Basin Observational System (NABOS) expedition during the late
summer of 2021 in the Arctic Ocean. A detailed water mass analysis was performed to clarify the potential sources of Fe and Mn in the surface LESS. Sea ice cores were also collected to calculate Fe and Mn sea ice inventories, and the potential supplies to the surface ocean upon melting were evaluated. Moreover, we investigated how dissolved organic matter interacts with Fe and Mn in the LESS. By combining these datasets, we interpreted the factors controlling the concentrations and distributions of Fe and Mn on the surface of the LESS.

## 2 Materials and Methods

### 2.1. Shipboard Sampling

Observations in the LESS were conducted onboard the Russian Research Vessel *Akademik Tryoshnikov* from September to October 2021 (Fig. 1a). Low-density polyethylene (LDPE) bottles and buckets (Thermo Fisher Scientific, USA), polyethylene bags (GL Sciences, Japan), Acropak capsules with Supor membrane filters (0.8/0.2-μm pore size, Pall, USA), and a Tygon
tube (Masterflex, Germany) used for sampling trace metals were thoroughly acid-cleaned in a class-100 clean-air laboratory. Seawater was collected from the side of the ship at a depth of approximately 10 m using a peristaltic pump (Geopump, Geotech Environmental Equipment, USA) and a Tygon tube. Samples of the dissolved fractions of Fe and Mn (dFe and dMn) were collected into LDPE bottles after filtration through Acropak filters connected to a Tygon tube. The samples for nutrient analysis were collected into acrylic vials after filtration. Samples for dissolved organic matter (DOM) were collected into pre-
combusted (450°C for 5 h) glass bottles after filtration through pre-combusted glass fiber filters (0.7-μm nominal pore size, Whatman GF/F, UK). The samples for total dissolvable Fe and Mn (TdFe and TdMn) and stable isotope of oxygen ($\delta^{18}O$) in the seawater were collected without filtration into LDPE bottles and glass bottles with rubber inserts in the caps, respectively. The pH for Fe and Mn samples was adjusted to < 1.8 by adding ultrapure grade 6 M hydrochloric acid (Tamapure AA-100, Tama Chemicals, Japan), and they were stored for a year before the analysis. The samples for nutrients and DOM were frozen
immediately after collection at −20°C and were shipped back to the onshore laboratory.

The water properties at depths of 0–30 m were measured using a portable CTD sensor (RINKO 102, JFE Advantech, Japan) (Fig. 2). Full-depth temperature and salinity profiles were obtained using a Seabird SBE911plus CTD system (Figs. S1 and S2).





**Figure 1** (a) Location of sampling stations during the 2021 NABOS expedition. Sea ice concentration (SIC, %) on 1 October 2021 is derived from GCOM-W/AMSR2 (Japan Aerospace Exploration Agency). General water flows on the surface of LESS are depicted according to Anderson et al. (2015), Bauch et al. (2018), Clement Kinney et al. (2022), Doglioni et al. (2022), Rudels et al. (2004), and Stabeno et al. (2018). Spatial distributions of (b) water temperature, (c) salinity, and (d) $\delta^{18}O$ at the depth of 10 m in LESS. Sea ice observations were conducted on 1st (St. Ice 1), 2nd (St. Ice 2), and 10th (St. Ice 3) October 2021. Abbreviations: Amundsen Basin (AB), Makarov Basin (MB), Chukchi Abyssal Plain (CAP), and East Siberian Sea (ESS).

## 2.2. Sampling on sea ice and sample processing

Sea ice observations were conducted at three ice stations from October 1 to 10, 2021 (Fig. 1a). Water sampling under sea ice was performed for analyses of Fe, Mn, nutrients, DOM, and $\delta^{18}O$ at depths of 1, 5, and 10 m using the pump system described in section 2.1. Sea ice cores were collected using an ice corer (Mark II Coring System, Kovacs Enterprise, USA) and sectioned



into five subsamples using a titanium flat-head screwdriver. The subsamples were cleaned onsite by removing more than 2 cm of their outer layers using acid-cleaned ceramic knives, according to previously reported methods (Kanna et al, 2014; Evans and Nishioka, 2019). The cleaned ice samples were then transferred to LDPE buckets. Snow samples were collected in polyethylene bags using an acid-cleaned polycarbonate scoop. The snow and cleaned ice samples were melted inside a class-
100 clean air bench placed onboard the laboratory. The meltwaters for analyses of Fe, Mn, nutrients, DOM, and $\delta^{18}O$ were subsampled and processed in the manner described in section 2.1. The salinity of the meltwater samples was measured using a portable salinity sensor (Multi 3510, Xylem, USA).

## 2.3. Sample analysis

The acidified water samples for Fe and Mn were first pre-concentrated using a manual solid-phase extraction system equipped
with a Nobias Chelate-PA1 resin column (Hitachi High Technologies, Japan) (Sohrin et al., 2008; Kondo et al., 2016). Ultrapure-grade nitric acid ($HNO_3$), acetic acid (HAc), and ammonium solution ($NH_3$) (Tamapure AA-100, Tama Chemicals, Japan) were used for pre-concentration and extraction. The water samples were adjusted to pH $6.0 \pm 0.1$ by adding 3.6 M ammonium acetate buffer solution prepared from HAc and $NH_3$. Fe and Mn concentrated on the resin were eluted with 2 M $HNO_3$ and analyzed using high-resolution inductively coupled plasma mass spectrometry (ELEMENT XR, Thermo Fisher
Scientific, USA). Procedure blanks of Fe and Mn were evaluated using ultrapure water following the preconcentration procedures, which showed $0.13 \pm 0.04$ nmol $kg^{-1}$ (n = 84) for Fe and $0.002 \pm 0.005$ nmol $kg^{-1}$ (n = 84) for Mn, respectively. Certified reference materials for trace metals, NASS-7 and CASS-6 (National Research Council of Canada), were measured to validate our pre-concentration procedures. The analytical values were within the error ranges of the certified reference materials (Table 1).
Dissolved organic carbon (DOC) was analyzed using a total organic carbon analyzer (TOC-L, Shimadzu, Japan). Absorbance spectra for chromophoric dissolved organic matter (CDOM) were analyzed between 200 and 800 nm at 1-nm intervals using a dual-beam spectrophotometer (UV-1800, Shimadzu, Japan) with a 1 cm quartz-windowed cell. The sample spectra were corrected using ultrapure water spectra and converted to Napierian absorption coefficients at wavelength ($a$ ($\lambda$), $m^{-1}$) (Green & Blough, 1994).
Excitation-emission matrix fluorescence spectra (EEMs) for the CDOM were analyzed using a fluorescence spectrophotometer (FP-8500, JASCO, Japan). EEMs were obtained at excitation wavelengths (Ex) ranging from 250 to 500 nm and emission wavelengths (Em) ranging from 280 to 600 nm. The fluorescence intensity was corrected to the area under the Raman peak of ultrapure water (Ex = 350 nm) and calibrated to Raman units (RU) using to the method outlined by Lawaetz and Stedmon (2009) and Tanaka et al. (2016).
The $\delta^{18}O$ value of the water samples was determined using an isotope water analyzer (Picarro L2120-i, Picarro, USA) with an analytical precision of ±0.3‰. Macronutrient concentrations were determined using an autoanalyzer (QuAAtro, BL TEC,



Japan) with a continuous flow system. The measurements were calibrated using reference seawater materials (KANSO Technos, Japan).

## 2.4. CDOM characterization

The CDOM absorption coefficient at 350 nm ($a_{350}$) was used as an indicator of terrestrial humic substances. In addition, parallel factor analysis (PARAFAC) was applied to statistically decompose the EEMs into their components. PARAFAC was performed in MATLAB (Mathworks, Natick, MA, USA) using the DOMFluor toolbox (Stedmon & Bro, 2008). The dataset used in this study comprises 42 samples of seawater, snow, and sea ice. The brine sample ($n = 1$) was assessed as an outlier in the PARAFAC model, and was not used in this study. The wavelength for PARAFAC was obtained at Ex ranging from 250

to 500 nm and Em ranging from 280 to 535 nm. The determination of the correct number of components was primarily achieved using split-half analysis and random initialization (Stedmon & Bro, 2008). The three-component model was validated based on the PARAFAC model (Fig. S3). Components 1 (C1) and 2 (C2) exhibit fluorescence peaks in the visible region and are defined as visible fluorescence. C1 peaked at an emission wavelength of 410 nm, which is traditionally categorized as a humic-like fluorophore of marine origin (Coble, 1996). C2 peaked at an emission wavelength of 470 nm, which is traditionally

categorized as a humic-like fluorophore of terrestrial origin (Coble, 1996), even though it has been reported that marine microbes produce this type of fluorophore (Goto et al., 2020). In this study, C1 was combined with C2 and interpreted as all humic-like fluorophores because their fluorescence intensities were not very different among the sampling locations. Component 3 (C3) exhibited fluorescence peaks in the ultraviolet A (UVA) region and is defined here as UVA fluorescence. This fluorophore is traditionally categorized as a protein-like fluorophore (Coble, 1996).

## 3. Results

### 3.1. Hydrography of LESS surface waters

Water properties in the ESS and CAP differed significantly from the MB and AB in the late summer of 2021. The surface waters in the ESS and CAP were relatively cold (temperature ($T$) < 0°C) and fresh (salinity ($S$) < 30) compared with those in the AB and MB (Figs. 1b, c, and S1). The waters in the ESS and CAP were also characterized by low $\delta^{18}O$ values of less than

−2.5‰ (Fig. 1d). The differences among regions were likely due to the magnitude of the mixing of Pacific-sourced water with Atlantic-sourced water as well as river runoff and melting of sea ice. The Pacific-sourced water enters from the Bering Strait, passes through the Chukchi Sea, and then penetrates the ESS (Fig. 1a). Atlantic-sourced water enters the AB and flows along the continental slope (Fig. 1a). The mixed layers in the Atlantic and Pacific sectors have different geochemical and physical characteristics. In the $T$ versus $S$ diagram, the properties of the upper water layer (< 30 m) in the ESS and CAP (Sts. 71, 89,

and Ice 1; Fig. 1b) were similar to those of Surface Polar Mixed Water (PMW) that originated in the Pacific sector of the Arctic Ocean, which is characterized by $T < 0$ and $S < 31$ (Fig. 2). In contrast, the upper water layer found in the AB and MB (Sts. 1,




11, 31, and Ice 3; Fig. 1b) is not derived from PMW but is a product of mixture of warm, saline Atlantic Water (AW) ($T > 0$ and $S > 34.8$) and freshwaters (Fig. 2). The $T$ of water above 25 m exceeded 0°C at 125−145 °E (Figs. S1 and S2), owing to the atmospheric radiative forcing.

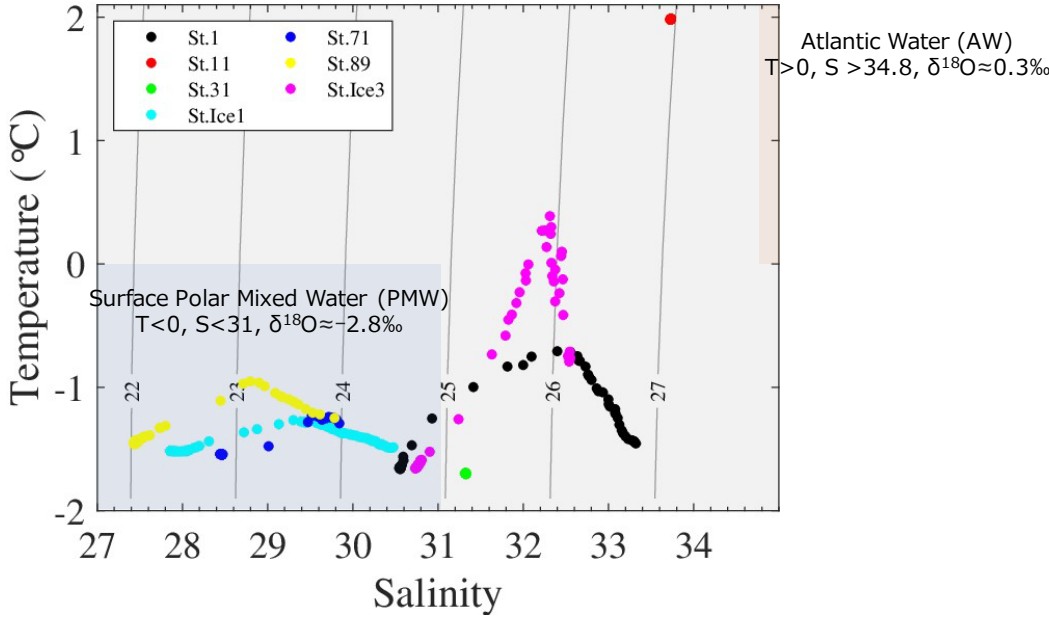


**Figure 2** Temperature versus salinity diagram at the 0−30 m depth along with isopycnals ($\sigma_t$) at certain stations in the LESS in 2021. The temperature and salinity ranges of Surface Polar Mixed Water (PMW) and Atlantic Water (AW) are indicated by blue- and red-hatched areas, respectively.

The surface water in the ESS and CAP was generally enriched in silicate (Si) and Mn, but depleted in Fe relative to the AB and MB (Figs. 3a−e). The concentrations of Si and TdMn reached 8.9 µmol L$^{-1}$ and 83.5 nmol kg$^{-1}$, respectively and the concentration of TdFe was as low as 0.9 nmol kg$^{-1}$ in the ESS and CAP. The TdFe concentration tended to be high along the continental slope (Figs. 3d and e), with a maximum value of 79.2 nmol kg$^{-1}$. Differences between the Fe and Mn distributions were also observed at two shallow (approximately 40 m bottom depth) southernmost stations on the shelf of ESS (Sts. 86 and

89; Fig. 3). The TdFe concentrations were relatively low (1.58−1.62 nmol kg$^{-1}$) at the stations where the notably high TdMn were detected (70.7−83.5 nmol kg$^{-1}$). The dFe and dMn, which do not include particles, showed spatial variations similar to those of TdFe and TdMn. The concentration differences between the unfiltered (i.e., TdFe and TdMn) and filtered samples (i.e., dFe and dMn) were attributed to the acid-labile particulate fraction of these metals. The acid-labile particulate fractions of Fe accounted for 48−61% of TdFe but only 6−18% of TdMn at the stations on the shelf. The results indicate that Fe was

relatively hosted in the particulate phase and Mn in the dissolved phase on the shelf. Various DOC concentrations were





observed at different sampling locations (Fig. 3f), and exceeded 100 μmol L$^{-1}$ at most stations. The detailed DOC distributions are discussed with optical information of CDOM in section 4.3.

**Figure 3** Spatial distributions of (a) Si, (b) dMn, (c) TdMn, (d) dFe, (e) TdFe, (f) DOC, (g) $f_{Pacific}$, (h) $f_{sim}$, and (i) $f_{mw}$ in the surface of LESS in 2021.

### 3.2. Water mass analysis

To quantify the relative contribution of freshwater to changes in the surface water properties of the LESS, we assumed a mixture of four components: meteoric water, sea-ice meltwater, Pacific Water, and Atlantic Water. Fractions of mass, salinity, δ$^{18}$O, and the nitrate to phosphate ratio (N/P) balance equations are described using the water properties of these four endmembers (Bauch et al., 2011; Charette et al., 2020; Gerringa et al., 2021; Newton et al., 2013):

$$f_{mw} + f_{sim} + f_{Pacific} + f_{Atlantic} = 1, \tag{1}$$





$$f_{mw} \cdot S_{mw} + f_{sim} \cdot S_{sim} + f_{Pacific} \cdot S_{Pacific} + f_{Atlantic} \cdot S_{Atlantic} = S, \tag{2}$$

$$f_{mw} \cdot \delta_{mw} + f_{sim} \cdot \delta_{sim} + f_{Pacific} \cdot \delta_{Pacific} + f_{Atlantic} \cdot \delta_{Atlantic} = \delta, \tag{3}$$

$$f_{mw} \cdot P_{mw} + f_{sim} \cdot P_{sim} + f_{Pacific} \cdot P_{Pacific} + f_{Atlantic} \cdot P_{Atlantic} = P, \tag{4}$$

where $f$, $S$, $\delta$, and $P$ are the fractions of mass, salinity, $\delta^{18}O$, and N/P-based phosphate concentration, and the suffixes mw, sim, Pacific, and Atlantic indicate meteoric water, sea-ice meltwater, Pacific Water, and Atlantic Water, respectively (Table 3). The measured value of nitrate+nitrite ($N$) in each sample were used to compute individual phosphate ($P$) endmembers for the

Pacific and Atlantic fractions from the Atlantic Water Line ($P = 0.0596 \times N + 0.1139$; Bauch et al., 2011) and Pacific Water Line ($P = 0.0653 \times N + 0.94$; Jones et al., 2008) for each sample. It should be noted that this N/P-based calculation produces slightly negative fractions of Pacific Water ($f_{Pacific}$) due to inaccuracies in endmembers and measurements (Bauch et al., 2011). Indeed, our calculation showed negative $f_{Pacific}$ at four stations with an average of $-2\%$. However, this error remains relatively small within the Atlantic regime and is still within the uncertainty ($\sim$10% for marine waters) of the method (Yamamoto-Kawai

et al., 2008).

The mass fractions of $f_{Pacific}$, sea ice meltwater ($f_{sim}$), and meteoric water ($f_{mw}$) components computed by solving Equations (1)–(4) are shown in Figures 3g−i. The results indicate that $f_{Pacific}$ in the surface water was greater than 50% in the ESS and CAP, whereas the fraction was less than 20% in the MB (Fig. 3g). Small values of $f_{Pacific}$ (up to 2%) are also found in the AB. At this location however, the $f_{Pacific}$ signal does not originate from Pacific-sourced water, but from modifications of water by

denitrification within the sediment over the Laptev Shelf (Bauch et al., 2011). A negative or positive value of $f_{sim}$ indicates the addition of brine or meltwater, respectively, to the surface layer (Fig. 3h). The $f_{sim}$ in the surface water was as high as 5.8% over the ESS and 3.4% over AB (Fig. 3h), where the melting of sea ice was likely predominant. In contrast, a negative value of $f_{sim}$ along the continental slope suggests that sea ice formation is dominant in the region, in agreement with previous work (Bauch et al., 2011; Yamamoto-Kawai et al., 2005). The $f_{mw}$ in the surface water increased from west to east by approximately

20% (Fig. 3i). Long-term trend of $f_{mw}$ is seen to increase in the Pacific sector of the Arctic Ocean since 1981 (Polyakov et al., 2020), which might be attributed to an influence from Siberian river runoff or increased freshwater flux through Bering Strait.

### 3.3. Physical and chemical properties of sea ice

During the NABOS expedition in the summer of 2021, snow, sea ice, and under-ice water were collected from three ice stations (Fig. 1). Detailed observations of the ice cores and snow pits at these stations can be found in the NABOS 2021 Cruise Report

(https://uaf-iarc.org/nabos-products/). Thin section analysis of a single ice core collected at St. Ice 2 revealed that the sea ice was composed of approximately 9% granular ice, 37% columnar ice, and 54% mixed ice, respectively (Fig. S4). Salinity and $\delta^{18}O$ values in the sea ice were similar among stations, ranging from 0 to 3.8 and from $-3.6$ to 0‰, respectively (Figs. 4a and b). Generally, Fe and Mn concentrations in the sea ice were lower than those in the under-ice water, except for the ice section (0.2 m from the bottom) collected at St. Ice 3 (Figs. 4c−f). The dFe concentrations in the sea ice gradually decreased from 3.2



to 1.7 nmol kg$^{-1}$ with the ice core depth, while the dMn concentrations increased from 2.4 to 5.2 nmol kg$^{-1}$. The TdFe and TdMn concentrations exhibited vertical variations similar to those of the dissolved fractions. The TdFe concentration tended to be higher in snow samples than in sea ice and under-ice water; however, this did not apply to TdMn.

We calculated the inventories of Fe and Mn from the cumulative metal loads of the 110 cm ice cores collected at St. Ice 1 (Table 2). The metal inventory of the ice core was comparable to that obtained from the Canada Basin (Evans and Nishioka, 235 2019), except for TdFe, which had a lower inventory. Moreover, the reported particulate Fe inventory in the ice core from the Western-Central Arctic Ocean (Bolt et al., 2020) was an order of magnitude exceeding the TdFe inventory in this study. Such discrepancies are likely due to greater heterogeneity in the distribution of particulate Fe loads in Arctic pack ice (Bolt et al., 2020). The sediment loadings of the collected ice samples were low based on visual observations, such that there was no evidence of ice-rafted sediment adding Fe and Mn to the surface waters, although their importance has been reported 240 (Hölemann et al., 1999a, 1999b; Measures, 1999; Rogalla et al., 2022; Tovar-Sánchez et al., 2010).

The DOC concentration varied in the snow (24.5−105 μmol L$^{-1}$) and sea ice samples (32.1−147 μmol L$^{-1}$), while the concentrations were relatively uniform vertically in the under-ice water (84−110 μmol L$^{-1}$) (Fig. 4g). The EEM results for CDOM showed that the intensities of visible fluorescence were lower in snow and sea ice than in under-ice water (Fig. 4h). The intensity of UVA fluorescence was relatively high in sea ice at St. Ice 2, but not at St. Ice 1 (Fig. 4i). Macronutrients were 245 generally depleted in the sea ice, except for nitrate+nitrite at St. Ice 2, which showed enrichment of 1.3 μmol L$^{-1}$ compared to the under-ice water (less than 0.25 μmol L$^{-1}$) (Figs. 4j−l). The enrichment of nitrate+nitrite was also observed in the snow samples (1.2 μmol L$^{-1}$).



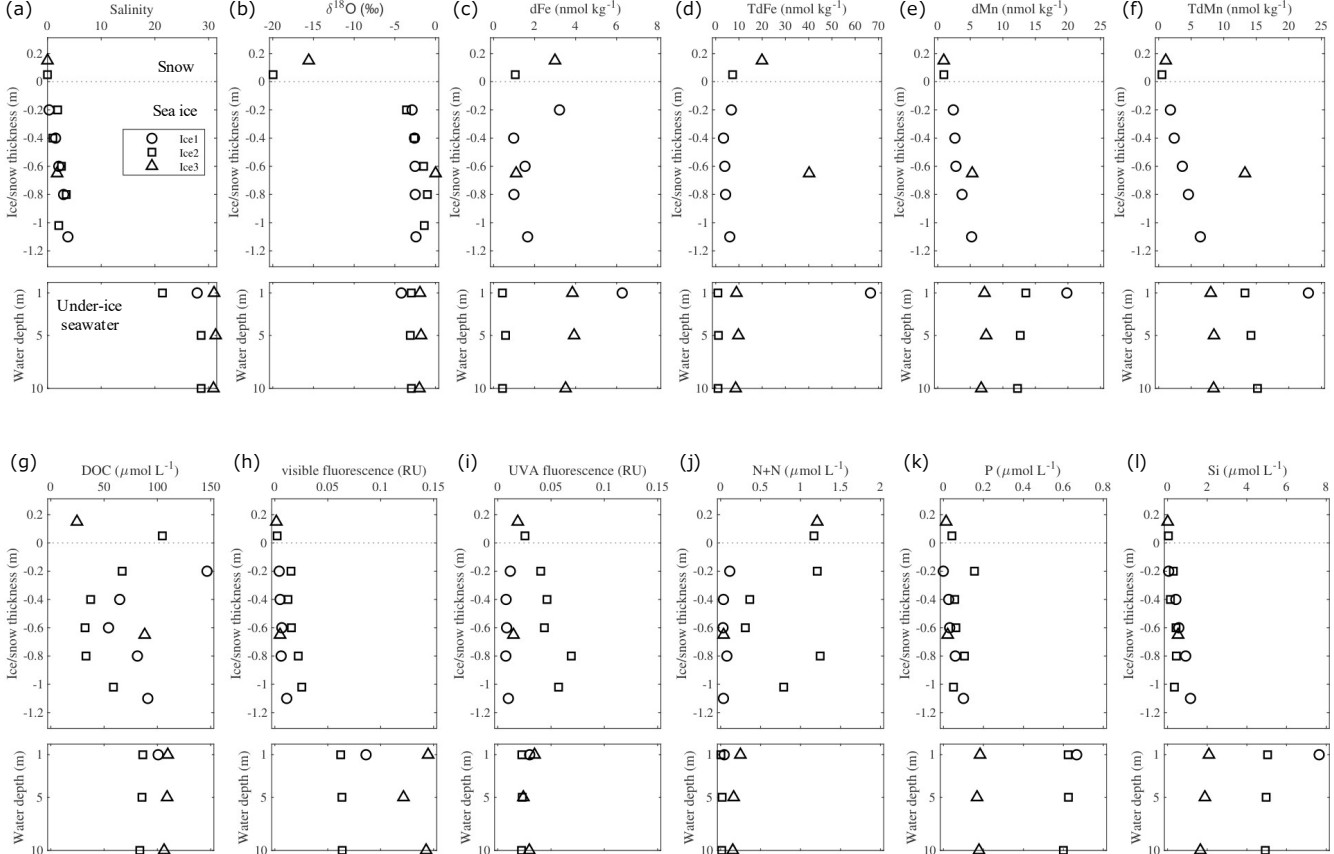

**Figure 4** Vertical profiles of (a) salinity, (b) δ¹⁸O, (c) dFe, (d) TdFe, (e) dMn, (f) TdMn, (g) DOC, (h) visible fluorescence, (i) UVA fluorescence, (j) nitrate+nitrite (N+N), (k) phosphate (P), and (l) silicate (Si) in snow, sea ice, and under-ice water, respectively.

## 4. Discussions

### 4.1. Comparison of Fe and Mn concentrations in seawater and freshwater sources

Variations of salinity and δ¹⁸O value in the surface water of LESS continental margins are attributed to local precipitation or river runoff and melting/formation of sea ice. Figure 5 shows the salinity versus δ¹⁸O diagram based on the samples of surface water, snow, sea-ice meltwater, and the previously reported freshwater sources (Evans and Nishioka, 2019; Marsay et al., 2019; Peterson et al., 2016). The salinity and δ¹⁸O values in most surface water samples deviated toward the meteoric endmember, including snow meltwater and the Siberian rivers. However, the salinity and δ¹⁸O values in the under-ice water at St. Ice 2 rather deviated toward the sea-ice endmember, resulting from substantial input of sea-ice meltwater into the station.




The dMn concentration in the surface waters gradually increased with decreasing salinity, whereas the dFe concentration did
       not show a similar increasing trend (Fig. 5). Previous studies have shown that notably high dFe and dMn concentrations in the
       Siberian rivers as having $1751 \pm 1218$ nmol L$^{-1}$ and $208 \pm 279$ nmol L$^{-1}$, respectively (Peterson et al., 2016), such that river
       waters must have supplied substantial amounts of Fe and Mn into the surface of LESS. In contrast, dFe and dMn concentrations
       in the sea-ice endmember were relatively low ($2.0 \pm 1.3$ nmol kg$^{-1}$ for dFe; $5.3 \pm 3.3$ nmol kg$^{-1}$ for dMn). The enriched dMn
in the under-ice water at St. Ice 2 (13.5 nmol kg$^{-1}$) was not likely derived from the input of sea ice meltwater because of the
       low level of dMn in the sea-ice endmember, suggesting the existence of extra Mn sources in the surface water. In the subsequent
       section, we discuss the factors controlling the distribution of Fe and Mn on the surface LESS.

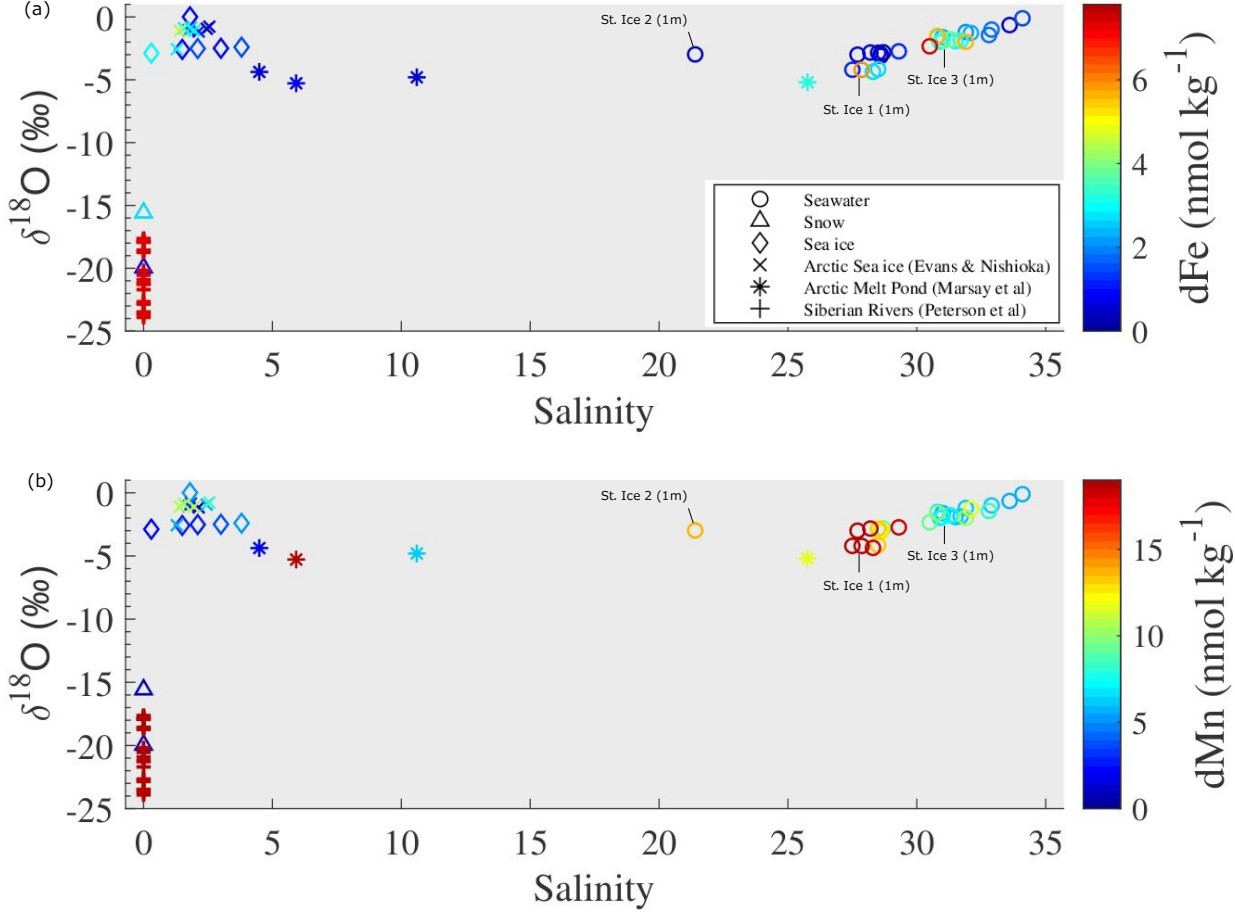

**Figure 5** Salinity versus δ$^{18}$O diagram in seawater, snow, and sea ice. The color scale shows concentrations of (a) dFe and (b)
dMn in each water sample. The values of other freshwater sources are cited from Evans and Nishioka (2019), Marsay et al.
       (2018), and Peterson et al. (2016).




## 4.2. Potential sources of Fe and Mn in the surface LESS

Based on the water mass analysis in section 3.2, we classified the surface water into PMW and Surface Atlantic Mixed Water (AMW) (Fig. 3g, dotted circle borders the area). The PMW contains a large fraction of Pacific-sourced water ($f_{Pacific} > 50\%$)
and is mostly found in the ESS and CAP, while the AMW is found in the AB and MB. The PMW is generally characterized by high Si, low Fe, and high Mn concentrations relative to AMW (Figs. 3a−e). The high Si content of the PMW likely results from nutrient-rich Pacific Water entering the shallow Bering Strait (Chen et al., 2018; Jensen et al., 2020; Nishino et al., 2013). The maximum surface Si concentration along the continental margin of the western MB, which is a typical signature of runoff from the Lena River in the region (Alling et al., 2010; Anderson et al., 2017), was not clearly observed in this study (Fig. 3a).
Fe and Mn are redox-active metals that share common sources in surface waters such as sediments, dust deposition, and freshwater inputs. The dust deposition is considered to be of minor importance in the LESS, given the relatively low concentrations of TdFe (~19.9 nmol kg⁻¹) and TdMn (~1.1 nmol kg⁻¹) observed in the snow samples in this study (Figs. 4d and f). In general, the summertime atmospheric deposition fluxes of Fe and Mn to the Arctic Ocean are reportedly low (Kadko et al, 2016; Marsay et al, 2018), especially in comparison to Arctic rivers and coastal erosion/diagenetic fluxes from shelf
sediments (Charette et al., 2020; Jensen et al., 2021; Kadko et al., 2018). In the following discussion, we evaluate the potential sources (sedimentary input, river runoff, and sea ice formation and melt) of these metals in PMW and AMW.

### 4.2.1. Sedimentary input

To distinguish shelf-derived Fe and Mn, parameter $N^*$ ($N^* = 0.87 \times (N − 16 \times P + 2.9)$; Gruber and Sarmiento, 1997) was evaluated for PMW and AMW. A negative or positive value of $N^*$ in water indicates a nitrate deficit (denitrification) or excess nitrates
(nitrogen fixation) relative to phosphate, respectively. In the Chukchi Sea, significant denitrification occurs within shelf sediments because nitrate is consumed instead of oxygen for organic matter decomposition (Yamamoto-Kawai et al., 2006). Nitrogen fixation also occurs throughout the Chukchi Sea; however, it is a minor process in the overall nitrogen cycle (Shiozaki et al., 2018). Negative $N^*$ could be an indicator of water passing through the reductive Chukchi Shelf and penetrating the MB (Nishino et al., 2013). Our results showed that the $N^*$ value in PMW was lower ($< −5$) than that in AMW (Fig. 6). The TdMn
concentrations tended to be high in low-$N^*$ PMW, suggesting a reductive sedimentary flux that released Mn from the Chukchi Shelf (Fig. 6b). Although Fe and Mn are thought to be released from reductive sediments over the Chukchi Shelf, TdFe concentrations in the PMW were relatively low (Fig. 6a) compared to those typically observed in the continental margin of the Arctic Ocean (Aguilar-Islas et al., 2013; Cid et al., 2011, 2012; Jensen et al., 2020; Klunder et al., 2012; Kondo et al., 2016; Nakayama et al., 2011; Nishimura et al., 2012). This is likely because Fe was much more rapidly removed in the Chukchi
Shelf water column than Mn via oxidation and re-precipitation (Jansen et al., 2020; Vieira et al., 2019; Millero et al., 1987). Indeed, we observed a lower dFe to TdFe ratio (43.6 ± 23.8) compared to dMn to TdMn ratio (85.2 ± 10.2) in PMW, such that Fe was primarily in the particulate phase and Mn was in the dissolved phase.





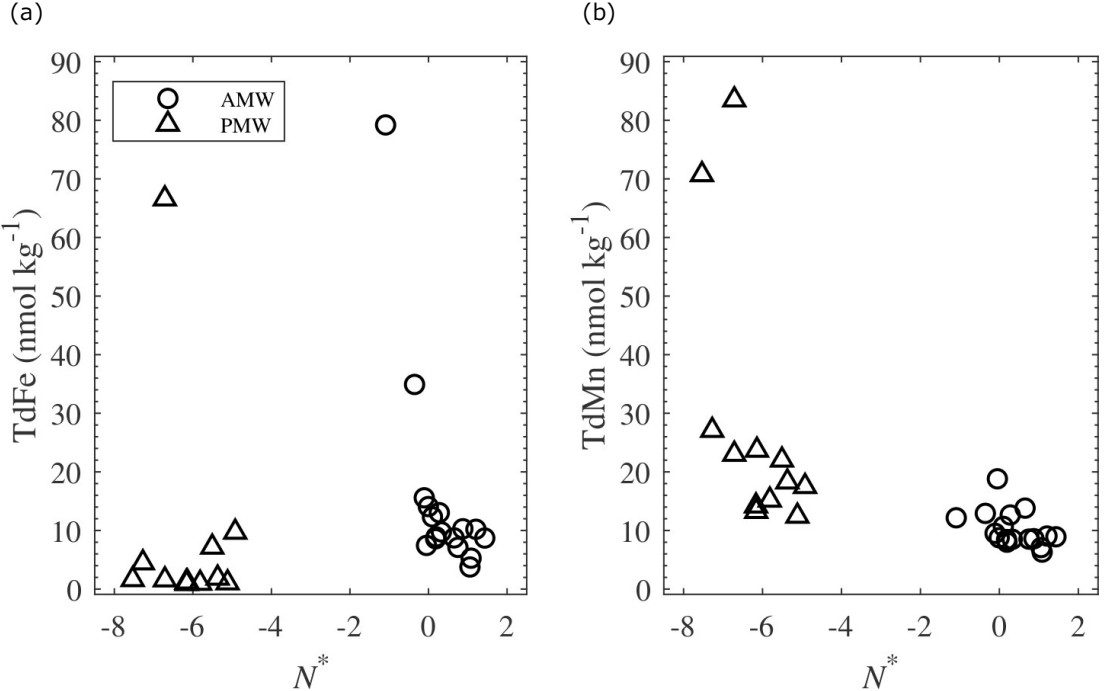

**Figure 6** Relationships between (a) TdFe and (b) TdMn concentrations and $N^*$ values in the AMW and PMW.

### 4.2.2. River runoff

$f_{mw}$ includes river runoff and local precipitation. The former dominates the signals of Fe and Mn sources in the Arctic surface waters (Dai and Martin, 1995; Guieu et al., 1996; Pokrovsky et al., 2016; Savenko and Pokrovsky, 2019). Although there was no clear relationship among $f_{mw}$, TdFe, and TdMn in the surface waters (Fig. S5), the dissolved fractions of the metals exhibited distinct characteristics. The dFe concentrations in the PMW and AMW were positively correlated with $f_{mw}$ (Fig. 7a), suggesting that river runoff is an important factor controlling dFe concentrations in surface waters. Three large Siberian rivers, the Lena, Kolyma, and Indigirka (Fig. 1a), discharge into the LESS, likely affecting the dFe concentrations on the surface waters. Moreover, Ob and Yenisei River waters flow into the northwestern Laptev Sea via eastward coastal current (Bauch et al., 2011, 2014). It should be noted that the contribution of river waters flowing into the Chukchi Sea (e.g., Yukon River) is already included within the Pacific-sourced water assignment based on the water mass calculations. Thus, the variation of $f_{mw}$ in PMW is mainly attributed to input from the Siberian rivers. At the same $f_{mw}$ level of about 10%, AMW had higher dFe relative to PMW. This could indicate the river sources contributing to AMW had a relatively higher dFe endmember or that some dFe in PMW underwent more removal processes than AMW. In contrast to dFe, the dMn concentrations in the PMW and AMW did not correlate to $f_{mw}$, individually (Fig. 7b). However, if we combined AMW with PMW in all water samples and considered outliers for samples that were largely influenced by sedimentary input (Sts. 86 and 89, Fig. 1b), the dMn concentrations in the



water were positively correlated with $f_{mw}$ ($r = 0.70$; $p < 0.001$; Fig. 7b). As a result, dMn exhibits characteristics that are different from those of dFe.

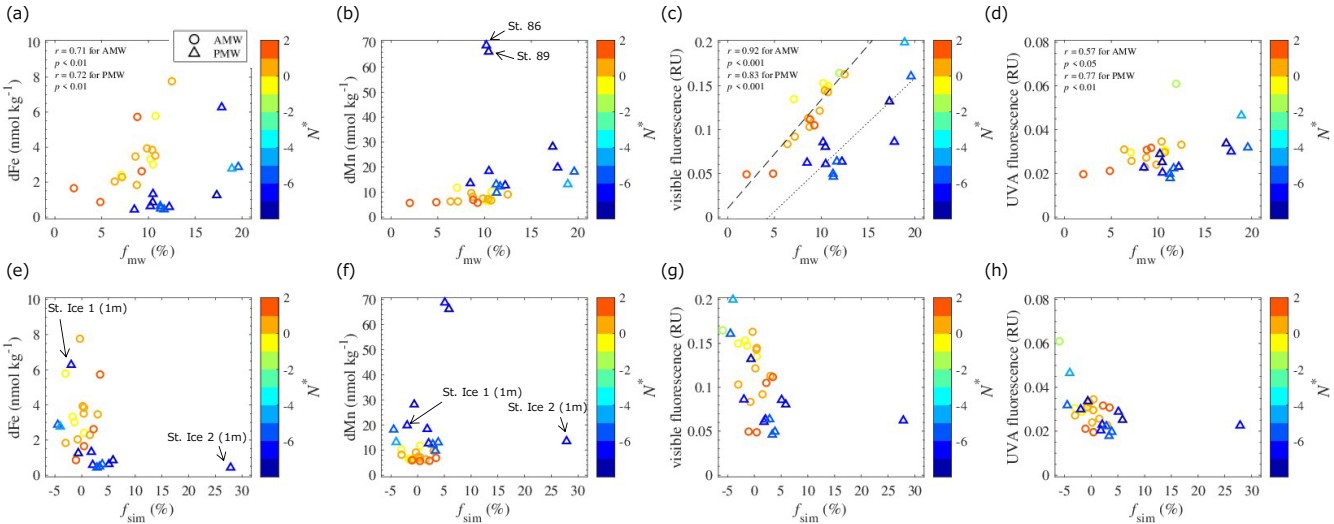

**Figure 7** Relationships between freshwater fractions ($f_{sim}$ and $f_{mw}$) and chemical components in the AMW and PMW. The color scale shows the $N^*$ values of each water sample. Linear relationships were evaluated based on Pearson correlation coefficients ($r$). The liner fits of the visible fluorescence-$f_{mw}$ relationships in the AMW and PMW are shown by dashed and dotted lines in (c), respectively.

Both metals are not equally preserved in seawater after being released from river water, owing to differences in removal rates. We examined the salinity-dMn/dFe ratio relationships among specific waters in the LESS (Fig. 8). Data of Yenisei River water and the estuarine water are also shown for the comparison. At Lena and Yenisei estuaries, the dMn/dFe ratio increases with increasing salinity because of the preferential loss of dFe relative to dMn after being released from these rivers (Fig. 8a and b). The dMn/dFe ratio in the AMW, as well as the surface water in the Laptev shelf/slope, was plotted around a mixing line of freshwater (rivers) and seawater (estuaries) (Fig. 8b). On the other hand, the dMn/dFe ratio in PMW extremely deviated from the freshwater-seawater line (Fig. 8a). This was due to the intrusion of dMn-excess shelf-derived water relative to dFe into the eastern part of the LESS, as discussed in section 4.2.1. Thus, the dMn distribution is driven by both input from shelf sediments and riverine flux, especially in the PMW.





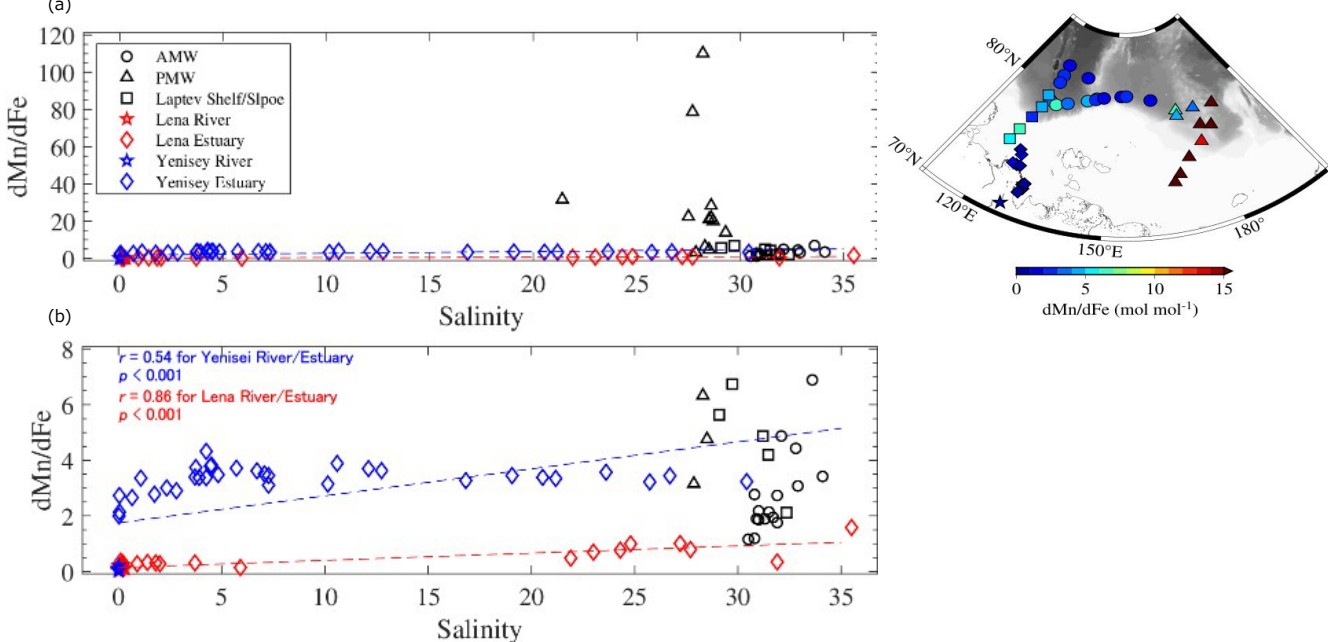

**Figure 8** (a) Relationships between salinity and dFe/dMn ratio in specific water mass. A smaller scale of the Y-axis is shown in (b). The geographical distribution of each sample is indicated by different symbols as shown in the map. Mixing lines of freshwater (Lena and Yenisei Rivers) and seawater (Lena and Yenisei estuaries) are derived from Hölemann et al. (2005), Peterson et al. (2016), and Savenko and Pokrovsky (2019). Linear relationships were evaluated against the freshwater-seawater line based on Pearson correlation coefficients ($r$). The value for surface water on the Laptev Shelf/Slope (Klunder et al., 2012; Middag et al., 2011) is plotted for comparison.

### 4.2.3. Sea ice formation and melt

345    The $f_{sim}$ includes brine drainage ($f_{sim} < 0$) or meltwater input ($f_{sim} > 0$) associated with sea ice formation or melting. The $f_{sim}$ values in PMW and AMW did not correlate with any metal concentration (Figs. 7e−f and S5). The $f_{sim}$ in the under-ice water at St. Ice 1, where we collected the ice core for trace metal analysis, showed a negative value of −2.0% (Fig. 7e−f), such that the water had received brine from the overlying sea ice. Indeed, the ice sample from St. Ice 1 was effectively permeable because the ice temperature was approximately −2℃ through the ice core with a bulk salinity of 2 (NABOS 2021 cruise report,

350    https://uaf-iarc.org/nabos-products/). For temperatures warmer than −5℃, brine-loading dissolved fractions of trace metals, as well as heat and nutrients, can move through the ice and finally be released to the water column (Golden et al., 1998; Lannuzel et al., 2008; van der Merwe et al., 2011). Given 2% of brine inclusion in the under ice-water at St. Ice 1, the added dFe and dMn into the water column associated with brine drainage account for 0.12 nmol kg$^{-1}$ and 0.39 nmol kg$^{-1}$, respectively. The



same calculation for the TdFe and TdMn showed 1.3 nM and 0.45 nM, respectively. The metal inputs throughout the brine
drainage were considered low compared to the other sources of Fe and Mn in the studied region.

In contrast to St. Ice 1, the under-ice water at St. Ice 2 was largely influenced by sea ice melt with $f_{sim}$ as high as 27.9% at a
depth of 1 m (Fig. 7e−f). The inventories of dFe and dMn over the depths of 1−10 m at St. 2 were computed as 4.5 μmol m$^{-2}$
and 115 μmol m$^{-2}$, respectively. The same calculation for the TdFe and TdMn showed 9.7 μmol m$^{-2}$ and 128 μmol m$^{-2}$,
respectively. The inventories of under-ice water were higher than those of the ice cores collected in this study (Table 3),
suggesting that sea ice melt is not the only source of these metals in under-ice water. It should be noted that this study did not
resolve the temporal evolution of sea ice melt, and thus we did not capture Fe and Mn concentrations in the sea ice at the early
stage of sea ice melt. A time series experiment conducted in the Antarctic pack ice demonstrated that 70% of Fe in the sea ice
was released into the under-ice water during the first 10 days of the survey, while the ice cover is still present (Lannuzel et al.,
2008). A lack of time series observation in this study may result in an underestimation of the inventories of metals in sea ice.
Nevertheless, sea ice formation and/or melting were less important in the overall Fe and Mn distribution in the surface LESS
during the study period, although the process had a potentially local impact on the Fe and Mn cycles.

An important conclusion in this section is that the major factors controlling Fe and Mn concentrations in the LESS were river
discharge and shelf sediment-water exchange processes. DOM is also sourced by rivers and seafloor sediment, which links to
the biogeochemical cycles of Fe and Mn in the Arctic (e.g., Charette et al., 2020). In the following section, we discuss the
linkage between DOM and these metals in the LESS continental margins.

## 4.3. Characteristics and origins of DOM and its relation to trace metals

In the surface LESS, various concentrations of DOC were observed with the highest value of 146 μmol L$^{-1}$ (Fig. 3f). Mean
value of DOC (108 ± 19 μmol L$^{-1}$) was similar to value previously observed in the Laptev Sea and ESS (Hölemann et al.,
2021) but tended to be high compared to the Chukchi Sea (Chen et al., 2018; Jung et al., 2021; Tanaka et al., 2016). In addition
to DOM release from productive shelf sediments (Cooper et al., 2005), regional DOM input from East Siberian rivers (Stedmon
et al., 2011) and inputs of fresh plankton-derived DOM (Davis and Benner, 2007) result in the surface waters of the LESS
having a high DOC signal. This study investigated the optical properties of CDOM, which are useful for understanding the
composition and origin of the DOM in the studied region.

The DOC concentrations in AMW and PMW were positively correlated with the absorption coefficient $a_{350}$ and the intensities
of visible fluorescence (Fig. 9a-b). This correlation implies that the factors controlling DOC concentrations and abundance of
humic-like CDOM are similar in these surface waters. The intensities of visible fluorescence in the AMW and PMW were
strongly correlated with $f_{mw}$ (Fig. 7c), suggesting the importance of riverine humic-like CDOM sources in surface waters.
Interestingly, the liner fits of the visible fluorescence-$f_{mw}$ relationships were different between AMW and PMW (Fig. 7c). This
result reflects that the riverine end-members of humic-like CDOM were different between AMW and PMW, i.e., Lena River





is the riverine end-member of AMW while Indigirka and/or Kolyma River is the end-member of PMW. The microbial processing of DOM can also generate visible fluorescence (Nelson et al., 2004; Rochelle-Newall and Fisher, 2002; Yamashita and Tanoue, 2008). The low-$N^*$ PMW must also have received microbe-mediated visible fluorescence from the benthic remineralization of sinking organic matter on the productive Chukchi Shelf (Hioki et al., 2014; Yamashita et al., 2019).

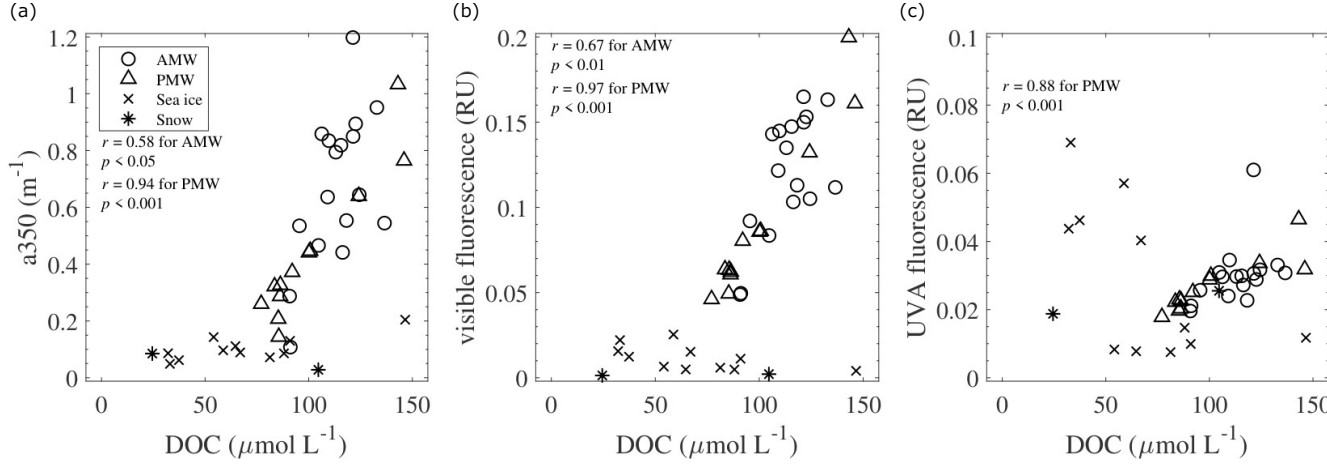


**Figure 9** Relationships between DOC and optical properties of DOM in AMW, PMW, sea ice, and snow. Linear relationships were evaluated based on Pearson correlation coefficients ($r$).

In addition to visible fluorescence, the intensity of UVA fluorescence in PMW was positively correlated with the DOC
concentration, but this did not apply to AMW (Fig. 9c). In the Arctic Ocean, UVA fluorescence is regarded as a phytoplankton-derived component (Brogi et al., 2019), including its direct release from phytoplankton and its release during grazing by zooplankton. We integrated the chlorophyll fluorescence in the water column (0–10 m depth) of the PMW and compared it with UVA fluorescence intensity (Fig. S6). The linear correlation between chlorophyll fluorescence and UVA fluorescence in PMW ($r = 0.80$; $p < 0.01$) indicated that the local production (and degradation) of the UVA fluorescence is an important
process. In addition, the intensities of UVA fluorescence in PMW and AMW increased with increasing $f_{mw}$ (Fig. 7d). This result can be interpreted as the rapid transport of freshly produced labile CDOM by river water to the surface of the LESS.
The relationship between $f_{mw}$ and trace metal concentrations revealed that the riverine source of dFe behaved more conservatively than that of dMn (Fig. 7a-b), which could be explained by the existence of organic ligands complexed with Fe. Previous studies on the Arctic Ocean reported that the CDOM pool contains Fe-binding organic ligands in the form of humic
substances (Laglera et al., 2007, 2011; Laglera and van den Berg, 2009; Slagter et al., 2017; Williford et al., 2021). Complexes of Fe-humic substances account for approximately 80% of dFe concentrations in the Arctic Ocean, and concentrations of these complexes are highly correlated with CDOM as well as dFe concentrations (Laglera et al., 2019). Our results showed significant correlations between dFe and visible fluorescence both in the AMW and PMW (Fig. 10a). The inputs of humic





substances from the Siberian rivers in those waters were evident from the elevated intensities of visible fluorescence with $f_{mw}$

(Fig. 7c). Thus, humic substances strongly affect the dFe concentration in the seawater by complexation with Fe, which stabilizes Fe in the dissolved phase. A difference of the liner fits of dFe-visible fluorescence relationships between AMW and PMW (Fig. 10a) may be explained by the contribution of excess humic ligands from rivers to PMW. In addition to Fe, Mn is known to form organic complexes with the degradation products of organic matter, such as humic materials (e.g., Oldham et al., 2017) and biogenic siderophores (e.g., Parker et al., 2004). However, the complexation of organic ligands with Mn in the

Arctic Ocean is not well known. None of the AMW or PMW samples showed a significant correlation between dMn and visible fluorescence (Fig. 10b). Unlike dFe, dMn is not stabilized by humic-type organic complexation in surface waters.

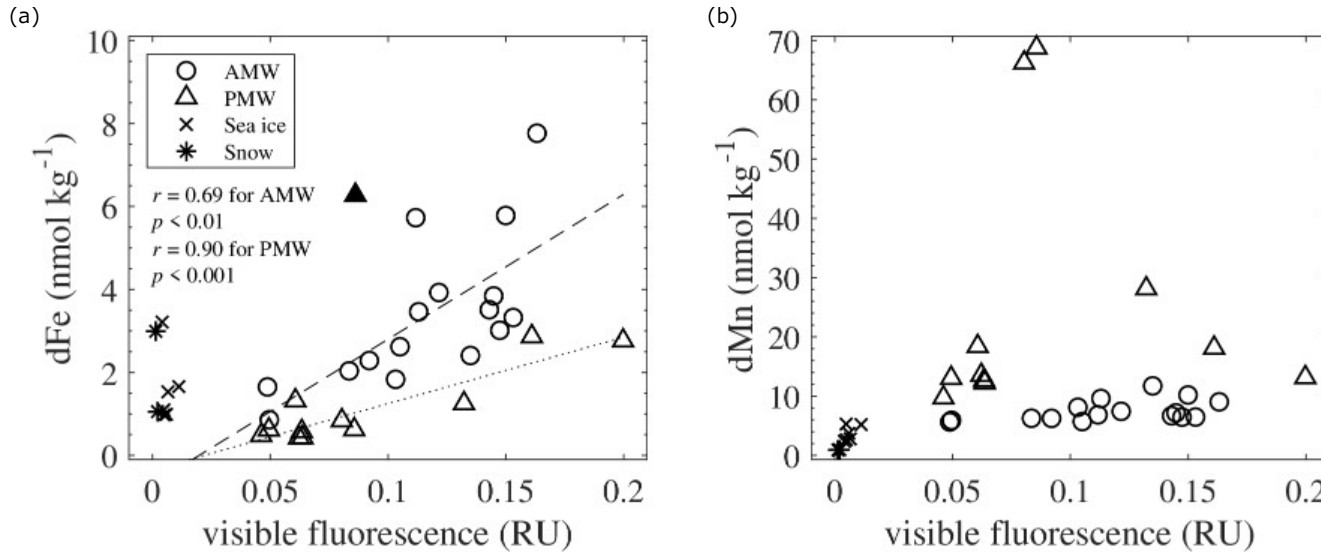

**Figure 10** Relationships of visible fluorescence with (a) dFe and (b) dMn in AMW, PMW, sea ice, and snow. Linear relationships were evaluated based on Pearson correlation coefficients ($r$), except for an outlier (▲) in (a). The liner fits of the

relationships in the AMW and PMW are shown by dashed and dotted lines in (a), respectively.

### 4.4. Comparison of Fe and Mn in LESS with other regions of the Arctic

To investigate the distribution patterns of dFe and dMn in surface waters, we combined our dataset with available data reported for the Arctic Ocean (Cid et al., 2011, 2012; Gerringa et al., 2021; Hölemann et al., 2005; Jensen et al., 2020; Klunder et al., 2012; Kondo et al., 2016; Middag et al., 2011; Rijkenberg et al., 2018; Savenko and Pokrovsky, 2019). The dFe concentrations

are relatively low (up to 2 nmol kg$^{-1}$) in the Atlantic sectors of the Arctic Ocean (Fig. 11a). In the surface of the Nansen Basin and Barents Sea, Fe is expected to be the first nutrient to be depleted by primary producers (Rijkenberg et al., 2018), and phytoplankton consumption could be an important sink for Fe. In the LESS continental margins, however, surface dFe concentrations are significant (>5 nmol kg$^{-1}$) and even persist in the late summer of 2021 (Fig. 11a). Other studies determined





that dFe concentration in the estuary of the Lena River was as high as 9,000 nmol kg$^{-1}$, as well as the estuaries of Yenisei and
Mackenzie Rivers (Fig. 11a). Fe-binding organic ligands in the form of humic substances originating from Lena River strongly
affect the dFe concentration here, as discussed in the previous section. The natural humic substances Fe ligands of the surface
Arctic Ocean have known to belong to the group of strong ligands ubiquitous in surface ocean waters (Laglera et al., 2019).
The strongly complexed Fe may be less biologically available to the phytoplankton community than the weakly complexed Fe
released from grazing and bacterial remineralization of organic matter (Gledhill and Buck, 2012). The river-influenced water
from the LESS continental margins is the source water of the Trans Polar Drift, which enriches in dFe in the central Arctic
Ocean (Fig. 11a) (Charette et al., 2020; Gerringa et al., 2021; Klunder et al., 2012).

High dMn found in the central Arctic Ocean is also related to the presence of the Trans Polar Drift (Charette et al., 2020;
Gerringa et al., 2021). In addition to the riverine inputs, sediment-water column exchange over the shelves leads to relatively
dMn-rich water in the Pacific sectors of the Arctic Ocean. The dMn concentrations increased toward the broad shelves of the
ESS (~68 nmol kg$^{-1}$), Chukchi Sea (~45 nmol kg$^{-1}$), and Bering Sea (~103 nmol kg$^{-1}$), whereas the concentration was relatively
low (~8 nmol kg$^{-1}$) over a narrow shelf of Beaufort Sea (Fig. 11b). As previously discussed, the shelf sediment-water exchange
processes over the Chukchi Shelf largely influence the Fe and Mn distributions in the ESS. Vieira et al. (2019) provided a first
estimate of benthic flux of the radium isotope (Ra$^{228}$) in the Chukchi Sea as tracers of benthic trace metal inputs, which was
among the highest rates reported globally. The low-$N$* water spreads over regions where nitrate is already depleted relative to
phosphate, mainly due to the oxidation of organic matter by bacteria in the reductive shelf sediment (Fig. 11c). The ESS, as
well as Chukchi Sea, are likely hotspots of sediment-sourced dMn via the reductive dissolution of Mn oxide in the sediment.
A multi-step removal process of dMn has been suggested in the Arctic (Jansen et al., 2020): dMn is rapidly removed to the
particulate phase within 150 km of the shelf break, but some dMn remains conserved through the next 1000 km away from the
shelf. The dMn originating from the LESS continental margins appears to be exported by the Trans Polar Drift to the central
Arctic Ocean effectively (Fig. 11b), even though stabilization by organic complexation is unlikely for Mn during offshore
transport.



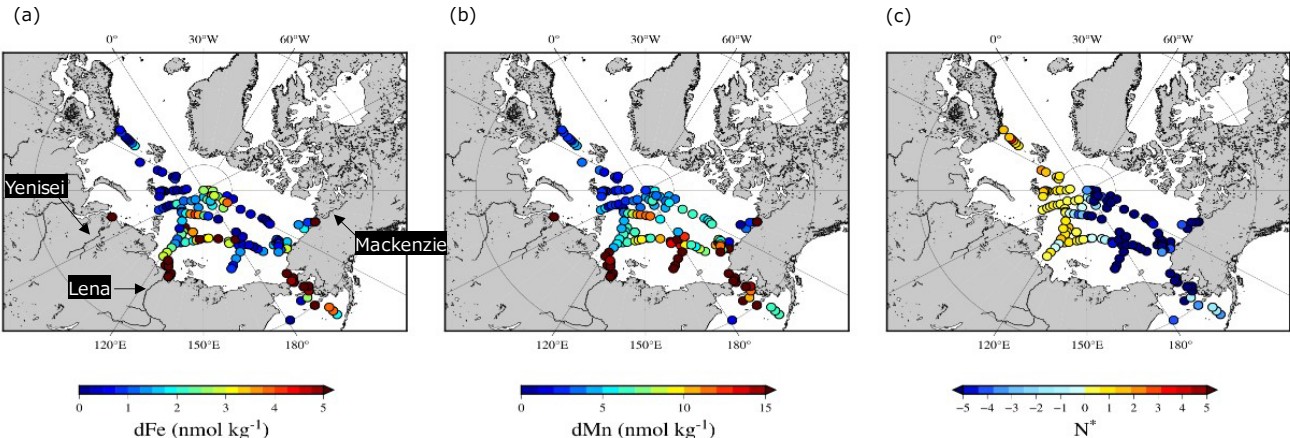

**Figure 11** Spatial distributions in (a) dFe, (b) dMn, and (c) $N^*$ in the Arctic Ocean.

## 5. Conclusion

Our results indicate that two governing hydrographic regimes exist in the surface waters of LESS continental margins in the late summer of 2021. The ESS and CAP are dominated by Pacific-sourced water, whereas AB and MB are dominated by the intrusion of Atlantic-sourced water in agreement with the previous works (e.g., Bauch et al., 2011). In these regions, the impact of river water discharge on the chemical properties of surface water is significant. However, sea ice melting/formation was less important during our observations. A positive correlation between the $f_{mw}$, dFe, and humic-like CDOM in the LESS

confirms a common freshwater source for dFe and humic-like CDOM. The humic-like organic ligands likely stabilized Fe in the dissolved phase, which was not the case for Mn. The ESS and CAP is characterized by particularly low $N^*$, resulting from a large sedimentary flux that releases Mn over the continental shelf. The LESS is a key region that originates from large amounts of shelf-derived nutrients, organic carbon, and trace elements (Charette et al., 2020; Jensen et al., 2021; Kipp et al., 2018). Shelf-derived materials as well as materials from riverine sources are transported to the central Arctic via the Trans

Polar Drift. Changes in the LESS region may affect the magnitude of material flux to the remote open ocean. This effect likely has a major impact on primary production and species composition in the Arctic surface waters. Further investigations in the LESS are required to elucidate how shelf- and river-derived elements are mixed within the water column and transported off the shelf.

## Acknowledgments

This expedition was supported by the ArCSII (JPMXD1420318865) International Exchange Program "Arctic Ocean: Improving tools and information for northern populations and safe navigation" and was coordinated by T. Alekseeva of AARI,



Russia. We thank the members of the 2021 NABOS expeditions. We are grateful to K. Kurashima and S. Otosaka for assistance with laboratory work. This research was funded by the Japanese Ministry of Education, Culture, Sports, Science and Technology through JSPS KAKENHI (20J01213, 20K19949 and 23K17028). This research was supported by a Grant for Joint
Research Program of the Japan Arctic Research Network Center.

**Code/Data availability**: The data are available upon request to the corresponding author (N. Kanna).

**Author contribution**: N. Kanna and H. Obata designed this study. I. Polyakov and T. Waseda supervised the NABOS expedition. N. Kanna, K. Tateyama, A. Timofeeva, M. Papadimitraki, and L. Whitmore carried out sampling, and N. Kanna
analysed samples. D. Nomura, H. Ogawa, and Y. Yamashita supervised $\delta^{18}O$, nutrients, and DOM analyses. All authors contributed to the manuscript.

**Competing interests**: The authors declare that they have no competing interests.

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



**Table 1** Results of measurement of certified reference materials for the trace metals, NASS-7 and CASS-6.

| Element | | NASS-7 ($\mu g\ kg^{-1}$) | | | CASS-6 ($\mu g\ kg^{-1}$) | |
| --- | --- | --- | --- | --- | --- | --- |
| | | Analytical value | Certified value | | Analytical value | Certified value |
| | $n$ | average ± sd | average ± sd | $n$ | average ± sd | average ± sd |
| Fe | 4 | 0.366 ± 0.018 | 0.344 ± 0.026 | 4 | 1.47 ± 0.01 | 1.53 ± 0.12 |
| Mn | 4 | 0.75 ± 0.004 | 0.74 ± 0.06 | 4 | 2.15 ± 0.02 | 2.18 ± 0.12 |


**Table 2** Comparison of inventories of Fe and Mn ($\mu mol\ m^{-2}$) in sea ice core from the Arctic.

| Samples | dFe | dMn | TdFe | TdMn | pFe | pMn | Reference |
| --- | --- | --- | --- | --- | --- | --- | --- |
| East Siberian Arctic Seas | 1.6 | 3.3 | 4.8 | 3.7 | – | – | This study |
| Canada Basin | 2.7 | 5.5 | 137 | 4.8 | – | – | Evans and Nishioka, 2019 |
| Western-Central Arctic | – | – | – | – | 11 to 63 | 0.4 to 1.5 | Bolt et al., 2020 |

TdFe (and TdMn) was determined in acidified unfiltered samples.

pFe (and pMn) was determined in acid-digested particle samples.

**Table 3** Endmember values of meteoric water, sea-ice meltwater, Pacific water, and Atlantic water for salinity, $\delta^{18}$O, and N/P-
based phosphate used in this study. The endmember value of sea-ice meltwater was obtained from this study, and others were
obtained from the studies performed by Bauch et al. (2011), Gerringa et al. (2021), Jones et al. (2008), and Newton et al.
(2013).

| Endmember | Salinity | $\delta^{18}$O (‰) | N/P-based phosphate ($\mu mol\ kg^{-1}$) |
| --- | --- | --- | --- |
| Meteoric water ($f_{mw}$) | 0 | –20 | 0.1 |
| Sea-ice meltwater ($f_{sim}$) | 2.1 | –2.08 | 0.06 |
| Pacific water ($f_{Pacific}$) | 32.7 | –1.1 | $0.0653 \times N + 0.94$ |
| Atlantic water ($f_{Atlantic}$) | 34.92 | +0.3 | $0.0596 \times N + 0.1139$ |