# Peer review of "Spatial distributions of iron and manganese in surface waters of the Arctic's Laptev and East Siberian seas"

_EGUsphere, 2024_

## Referee Comment (RC1)

**Review Report**

**Spatial distributions of iron and manganese in surface waters in the Arctic's Laptev and East Siberian seas**

Naoya Kanna , Kazutaka Tateyama , Takuji Waseda , Anna Timofeeva, Maria Papadimitraki , Laura Whitmore, Hajime Obata1, Daiki Nomura, Hiroshi Ogawa, Youhei Yamashita and Igor Polyakov

**Summary of the Study**

A descriptive study examining the mechanisms regulating the distribution of iron (Fe) and manganese (Mn) trace metals in the Arctic's Laptev and East Siberian Seas based on in-situ measurements. Overall, they find that river discharge and shelf sediment-water exchange processes are essential sources of Fe and Mn in this region. Contrasting the East Siberian Sea (ESS) and Chukchi Abyssal Plain (CAP) with the Makarov Basin (MB) and Amundsen Basin (AB), they find significant differences driven by source waters and biological activity. Surface waters in the ESS and CAP are generally enriched in Mn but depleted in Fe compared to those in the AB and MB. The ESS receives nutrients and trace metals from Pacific waters, in contrast to the Atlantic waters found in the MB and AB. Biological control of Fe through ligand complexation is a key mechanism, resulting in Fe predominantly existing in a particulate phase while Mn remains in the dissolved phase on the shelf.

**Major comments**

The in-depth characterization of the Arctic's Laptev and East Siberian Seas provided here is an essential contribution to understanding the biogeochemical and physical processes controlling trace metals budgets, especially given the recent accelerating changes in the region. The authors present a compelling mechanistic synthesis, but the manuscript requires refinement in its presentation. Overall, the manuscript is generally grammatically clean. However, it needs improvements in structure and readability. First, the manuscript reads more like a report or book chapter, overly descriptive and lacking a clear hypothesis, or answering specific questions. Though a valuable contribution, it is unclear why readers should read it. For instance, focusing on contrasting the physical and biogeochemical controls of iron (Fe) and manganese (Mn) in the Arctic's Laptev and East Siberian Seas could provide a better framework for the study. Second, the labeling of the figures is inadequate; nearly all figures lack titles, necessitating readers to refer to figure descriptions for context. Moreover, the figure descriptions do not adequately explain the terms used, and there is a notable presence of technical jargon throughout the study. Third, the manuscript overuses acronyms, which can hinder readability. Finally, some sentences contain incomplete reasoning, which detracts from the overall coherence of the manuscript. Below, I highlight few examples.

**Minor comments**

Line 165: Too many acronyms already, you don't need S and T for temperature and salinity.
Line 165: "The T of water above 25 m exceeded 0°C at 125−145 °E (Figs. S1 and S2), owing to the atmospheric radiative forcing" It is not clear what you mean in this sentence. What does radiative

forcing have to do with surface ocean temperature? These two concepts are obviously related, but it is unclear how you are using them

Line 190: Define the terms in your figures.
Line 215: "In contrast, a negative value of $f_{sim}$ along the continental slope suggests that sea ice formation is dominant in the region," which region?

Figure 4: Same issue with Figure 3, but you sometimes use full names. Be consistent..

Line 255: {The dMn concentration in the surface waters gradually increased with decreasing salinity, whereas the dFe concentration did not show a similar increasing trend (Fig. 5)} This statement is not a complete interpretation of figure 5, can you explain the exceptions

---

## Author Comment (AC1)

Response to Anonymous Referee #1 comments (RC1)

Major comments

**1)** RC1 The in-depth characterization of the Arctic's Laptev and East Siberian Seas provided here is an essential contribution to understanding the biogeochemical and physical processes controlling trace metals budgets, especially given the recent accelerating changes in the region. The authors present a compelling mechanistic synthesis, but the manuscript requires refinement in its presentation. Overall, the manuscript is generally grammatically clean. However, it needs improvements in structure and readability.

We are grateful for evaluating our manuscript and encouraging us to improve the manuscript. We tried improving the manuscript's readability after receiving the reviewer's comments.

First, the manuscript reads more like a report or book chapter, overly descriptive and lacking a clear hypothesis, or answering specific questions. Though a valuable contribution, it is unclear why readers should read it. For instance, focusing on contrasting the physical and biogeochemical controls of iron (Fe) and manganese (Mn) in the Arctic's Laptev and East Siberian Seas could provide a better framework for the study.

In response to the reviewer's comment, we revised the abstract to give a clearer hypothesis for this study and explain how we tackled our research questions. The revised abstract is as follows:

> "**Abstract.** The Arctic's Laptev and East Siberian Seas (LESS) are areas with high biogeochemical activity. Nutrient inputs associated with river runoff and shelf sediment-water exchange processes are vital for supporting primary production in the LESS. Relative to macronutrients, data on dissolved iron (dFe) and manganese (dMn), essential micronutrients for primary producers, have been historically sparse in the LESS. Some dFe and dMn are reportedly carried in the Central Arctic by the Transpolar Drift, a major current that directly transports Eurasian shelf water, river waters, and sea ice from the LESS continental margins. However, dFe and dMn supply to the surface waters of the LESS and subsequent biogeochemical processes are not well constrained. In the summer of 2021, we investigated the questions: *what are the sources of dFe and dMn to the surface layer* and *what factors control their concentrations and distributions on the LESS continental margins*? We demonstrated strong regional controls in dFe and dMn distribution based on distinct hydrographic regimes between the eastern side of the LESS (the East Siberian Sea and the Chukchi Abyssal Plain) and the western side (the Makarov and Amundsen Basins). Specifically, the East Siberian Sea and Chukchi Abyssal plain were governed by Pacific-sourced water and the Makarov and Amundsen Basins were influenced by Atlantic-sourced water. Pacific-sourced waters contained higher levels of dMn released from the continental shelf sediments than Atlantic-sourced water. In contrast, elevated dFe signals were not observed; which is likely because sedimentary dFe was more rapidly removed from the water column through oxidation or scavenging than dMn. The impact of river water discharge on the dFe distributions of Pacific- and Atlantic-sourced water

is significant. A positive correlation between the fraction of meteoric water (river water and precipitation), dFe, and humic-like colored dissolved organic matter (CDOM) in these waters confirmed a common freshwater source for dFe and CDOM. Terrigenous organic ligands likely stabilized Fe in the dissolved phase, which was not the case for Mn. Sea ice melting/formation was not a significant source during our observation period. We conclude that the major sources controlling dFe and dMn distributions in the LESS continental margins are river discharge and the input of shelf sediment. "

We had already focused on contrasting the physical and biogeochemical controls of Fe and Mn in the LESS throughout the manuscript. For example, we discussed preferential removal of Fe relative to Mn with $N^*$, an indicator of water passing through the reductive Chukchi Shelf. We further added a summary figure and some discussion in section 4.4. to explain how Fe and Mn data from this study compares with the previous study. We are sure that we responded well with this discussion for the reviewer's comment. The added discussions and the figure are as follows:

Section 4.4.

[revised manuscript text omitted]

Second, the labeling of the figures is inadequate; nearly all figures lack titles, necessitating readers to refer to figure descriptions for context. Moreover, the figure descriptions do not adequately explain the terms used, and there is a notable presence of technical jargon throughout the study.

We have carefully and thoroughly revised the figures and their captions. All figures were provided titles for conciseness, and their descriptions were strived for clarity. Please check the revised all figures and their captions at the end of this file.

Third, the manuscript overuses acronyms, which can hinder readability.

Following the reviewer's indications, we avoided using acronyms as much as we can throughout the text. For example, we decided not to use acronyms of Amundsen Basin, Makarov Basin, Chukchi Abyssal Plain, and East Siberian Sea in the text, which strongly hinder readability.

Finally, some sentences contain incomplete reasoning, which detracts from the overall coherence of the manuscript. Below, I highlight few examples.

We revised the text following the reviewer's comments below. Detailed responses to the reviewer's comments are as follows.

Minor comments

**2)** RC1 Line 165: Too many acronyms already, you don't need S and T for temperature and salinity.

We deleted them as the reviewer suggested.

**3)** RC1 Line 165: "The T of water above 25 m exceeded 0°C at 125−145 °E (Figs. S1 and S2), owing to the atmospheric radiative forcing" It is not clear what you mean in this sentence. What does radiative forcing have to do with surface ocean temperature? These two concepts are obviously related, but it is unclear how you are using them

We realized that this sentence had been over-interpreted. We did not measure atmospheric radiative forcing. Therefore, we removed this sentence from the text.

**4)** RC1 Line 190: Define the terms in your figures.

We defined the terms in the figure caption as the reviewer suggested. The revised figure and its caption are as follows:

[Figure]

"**Figure 3** Spatial distributions of (a) silicate, (b) dMn, (c) TdMn, (d) dFe, (e) TdFe, and (f) DOC in the surface of the Arctic's Laptev and East Siberian Seas. "

**5)** RC1 Line 215: "In contrast, a negative value of fsim along the continental slope suggests that sea ice formation is dominant in the region," which region?

We revised the sentence as follows:

Line 215:

"In contrast, a negative value of $f_{sim}$ in the Makarov Basin suggests that sea ice formation is dominant, in agreement with..."

**6)** RC1 Figure 4: Same issue with Figure 3, but you sometimes use full names. Be consistent.

We revised the terms in the figure following the reviewer's comment. We decided not to use 'Si', 'N+N', and 'P' because these terms were not used frequently throughout the manuscript. We also do not use the acronym of salinity (S). The revised figure and its caption are as follows:

[Figure]

"**Figure 5** Vertical profiles of (a) salinity, (b) δ¹⁸O, (c) dFe, (d) TdFe, (e) dMn, (f) TdMn, (g) DOC, (h) visible fluorescence, (i) UVA fluorescence, (j) nitrate+nitrite, (k) phosphate, and (l) silicate in snow, sea ice, and under-ice water, respectively. "

**7)** RC1 Line 255: {The dMn concentration in the surface waters gradually increased with decreasing salinity, whereas the dFe concentration did not show a similar increasing trend (Fig. 5)} This statement is not a complete interpretation of figure 5, can you explain the exceptions

In response to the comments from the reviewer, we revised the sentence as follows:

> Line 255:
>
> "The dMn concentration in the surface waters gradually increased with decreasing salinity (Fig. 6b). The dFe concentration also shows a similar increasing trend among the salinity ranges from 31 to 34 (Fig. 6a), however, the dFe concentration in less saline waters (salinity < 31) is irrespective of salinity variation. "

**Figure 1**

[revised manuscript text omitted]

---

## Author Comment (AC2)

Response to Anonymous Referee #2 comments (RC2)

This research investigates the concentrations and distributions of dissolved iron (dFe) and manganese (dMn) in the surface waters of the Laptev and East Siberian Seas (LESS) during the late summer of 2021. The study concludes that the concentrations of dFe and dMn in the LESS are primarily controlled by river discharge and shelf sediment inputs rather than by sea ice melt. The manuscript contributes important data to the relatively understudied area of trace metal dynamics in the Arctic. The descriptions of sampling and analyzing methods are solid. While this manuscript leans more toward descriptive analysis, it still offers valuable insights into the region's biogeochemical processes. I have several comments on this manuscript:

We sincerely appreciate the reviewer for helping to improve our manuscript. We thoroughly revised it following the reviewer's comments.

Major comments

**1)** RC2 Revise the Abstract: The abstract lacks clarity and does not efficiently summarize the entire article. The wording is awkward in places, such as in the phrase "Nutrient-rich Pacific-sourced water exists...," which does not clearly convey that this water mass is dominant in the region. The conclusion section is clearer and does a better job of summarizing the findings. Please rewrite or revise the abstract for better clarity and conciseness.

Following the reviewer's comments, we revised the abstract for clarity and conciseness as follows:

"**Abstract.** The Arctic's Laptev and East Siberian Seas (LESS) are areas with high biogeochemical activity. Nutrient inputs associated with river runoff and shelf sediment-water exchange processes are vital for supporting primary production in the LESS. Relative to macronutrients, data on dissolved iron (dFe) and manganese (dMn), essential micronutrients for primary producers, have been historically sparse in the LESS. Some dFe and dMn are reportedly carried in the Central Arctic by the Transpolar Drift, a major current that directly transports Eurasian shelf water, river waters, and sea ice from the LESS continental margins. However, dFe and dMn supply to the surface waters of the LESS and subsequent biogeochemical processes are not well constrained. In the summer of 2021, we investigated the questions: *what are the sources of dFe and dMn to the surface layer* and *what factors control their concentrations and distributions on the LESS continental margins*? We demonstrated strong regional controls in dFe and dMn distribution based on distinct hydrographic regimes between the eastern side of the LESS (the East Siberian Sea and the Chukchi Abyssal Plain) and the western side (the Makarov and Amundsen Basins). Specifically, the East Siberian Sea and Chukchi Abyssal plain were governed by Pacific-sourced water and the Makarov and Amundsen Basins were influenced by Atlantic-sourced water. Pacific-sourced waters contained higher levels of dMn released from the continental shelf sediments than Atlantic-sourced water. In contrast, elevated dFe signals were not observed; which is likely because sedimentary dFe was more rapidly removed from the water column through oxidation or scavenging than dMn.

The impact of river water discharge on the dFe distributions of Pacific- and Atlantic-sourced water is significant. A positive correlation between the fraction of meteoric water (river water and precipitation), dFe, and humic-like colored dissolved organic matter (CDOM) in these waters confirmed a common freshwater source for dFe and CDOM. Terrigenous organic ligands likely stabilized Fe in the dissolved phase, which was not the case for Mn. Sea ice melting/formation was not a significant source during our observation period. We conclude that the major sources controlling dFe and dMn distributions in the LESS continental margins are river discharge and the input of shelf sediment. "

**2)** RC2 Reorganize the structure. The method for calculating water mass fractions, described in lines 181–210, should be moved to Section 2 (Methods)

Following the reviewer's suggestion, we created a new subsection 2.5 in the Method section and moved the calculation for water mass fractions there.

**3)** RC2 Lines 211–220 do not present the mass fraction of Atlantic water. This information should be included. And its also not shown in Figure 3.

As the reviewer indicated, we included the mass fraction of Atlantic water ($f_{\text{Atlantic}}$) in the figure. We made a new figure regarding the results of mass fraction analysis because the previous Figure 3 contained much information. The conclusion of the text has not been changed by this revision. The revised figure is as follows:

[Figure]

"**Figure 4** Spatial distributions of fractional (a) Pacific Water ($f_{\text{Pacific}}$), (b) Atlantic Water ($f_{\text{Atlantic}}$), (c) sea ice meltwater ($f_{\text{sim}}$), and (d) meteoric water ($f_{\text{mw}}$) in the surface of the Arctic's Laptev and East Siberian Seas. Abbreviations in (a) and (b): Surface Polar Mixed Water (PMW) and Surface Atlantic Mixed Water (AMW). "

**4)** RC2 Section 4.2.1: The sedimentary contribution to dFe and dMn is not adequately discussed.

In response to the reviewer's comment, we plotted dissolved fractions of the metals with $N^*$ as well. We also added a discussion to the text regarding the sedimentary contribution to dFe and dM. The revised manuscript is as follows:

Section 4.2.1

"Our results showed that the $N^*$ value in PMW was much lower ($< -5$) than that in AMW (Fig. 7). Although Fe and Mn are thought to be released in the dissolved phase from reductive sediments over the Chukchi Shelf, these metals are gradually removed from the water column as the particulate phase (Jensen et al., 2020). The TdMn and dMn concentrations tended to be high in low-$N^*$ PMW, suggesting a reductive sedimentary flux that released Mn from the Chukchi Shelf (Fig. 7b and d). Mn was more elevated at shallow shelf stations 86 and 89, which were the most influenced by shelf inputs. Given that the dMn to TdMn ratio in PMW is as high as 85.2 ± 10.2%, Mn was primarily in the dissolved phase. On the contrary, the TdFe and dFe concentrations in PMW were relatively low (Fig. 7a and c) compared to those typically observed in the continental margin of the Arctic Ocean (Aguilar-Islas et al., 2013; Cid et al., 2011, 2012; Jensen et al., 2020; Klunder et al., 2012; Kondo et al., 2016; Nakayama et al., 2011; Nishimura et al., 2012). This is likely because Fe was much more rapidly removed in the Chukchi Shelf water column than Mn via oxidation and re-precipitation (Jansen et al., 2020; Vieira et al., 2019; Millero et al., 1987). Indeed, we observed a lower dFe to TdFe ratio (43.6 ± 23.8%) in PMW, such that Fe was primarily in the particulate phase. The relatively high Fe concentrations at stations Ice-1, 38, and 40 are likely attributable to riverine flux because these stations show relatively high fractional $f_{mw}$ (see discussion below). "

[Figure]

"**Figure 7** Plots of (a) TdFe, (b) TdMn, (c) dFe, and (d) dMn in Surface Atlantic Mixed Water (AMW) and Surface Polar Mixed Water (PMW) against $N^*$ values."

**5)** RC2 Line 334–335: This section is unclear. Although sedimentary inputs to dFe and dMn were discussed in Section 4.2.1, the correlations between TdFe, TdMn, dFe, and dMn should be more clearly explained.

In response to the reviewer's comment, we plotted the correlations between TdFe, TdMn, dFe, and dMn in the supplement figure S6. We also explained the correlations between them in the Section 4.2. The revised text is as follows:

Section 4.2.

"A significant correlation between dFe and dMn has been observed in the deeper waters (>3000 m) of the Amundsen and Makarov Basins because scavenging removal is the dominant process in the deep water masses (Klunder et al., 2012). In the surface water of the LESS, Fe was not correlated with Mn in the unfiltered and filtered fractions (Fig. S6). More factors, such as external input, influence the dFe and dMn distribution in the surface waters, leading to a disappearance of the Fe-Mn relationship. Moreover, enrichment in dMn compared to dFe (Fig. S6a) was observed in all sampled surface water, suggesting the importance of input fluxes of dMn or the preferential scavenging of dFe relative to dMn."

[Figure]

"Figure S6 (a) Plot of dMn in Surface Atlantic Mixed Water (AMW) and Surface Polar Mixed Water (PMW) against dFe. (b) Plot of TdMn against TdFe. The dashed line in (a) presents the relation of both metals found in the deeper waters (>3000 m) of the Amundsen and Makarov Basins, with a correlation: $[dMn] = (0.15 \times [dFe])/0.75$ (Klunder et al., 2012)."

**6)** RC2 Figure 6 and Line 297: What are the possible explanations for the outliers with high TdFe and TdMn? Are these stations heavily influenced by sedimentary inputs? Please address these points.

We highlighted the outliers in the figure. The outliers with high TdMn were obtained at two shallow southernmost stations on the shelf of the East Siberian Sea, and therefore the outliers were likely influenced by sedimentary inputs. For the TdFe, the sampled water seems to be influenced by the inputs of river runoff because these stations show relatively high fractional $f_{mw}$. We addressed the points as follows:

Line 297:

"Although Fe and Mn are thought to be released in the dissolved phase from reductive sediments over the Chukchi Shelf, these metals are gradually removed from the water column as the particulate phase (Jensen et al., 2020). The TdMn and dMn concentrations tended to be high in low-$N^*$ PMW, suggesting a reductive sedimentary flux that released Mn from the Chukchi Shelf (Fig. 7b and d). Mn was more elevated at shallow shelf stations 86 and 89, which were the most influenced by shelf inputs. Given that

the dMn to TdMn ratio in PMW is as high as 85.2 ± 10.2%, Mn was primarily in the dissolved phase. On the contrary, ... The relatively high Fe concentrations at stations Ice-1, 38, and 40 are likely attributable to riverine flux because these stations show relatively high fractional $f_{mw}$ (see discussion below). "

**7)** RC2 Section 4.4 and Figure 11: The comparison with other studies is valuable, but Figure 11 caption is not informative enough and needs to be modified. Please properly cite data sources in the figure captions. Use different symbols for different data sources instead of the same symbols, and highlight the data from this study using distinct symbols. Additionally, I suggest adding a summary figure by region to show how the data from this study compares with or complements the range of data reported in previous research.

When we use different symbols on the figure with a color ramp, the figure looks hard to see. Instead, we revised the figure with proper symbols and citations as follows:

[Figure]

"**Figure 12** Spatial distributions in (a) dFe, (b) dMn, and (c) $N^*$ in surface water (< 25 m depth) in the Arctic Ocean. (d) Location of stations on dFe, dMn, and $N^*$ reported by this study and the previous studies."

It should be noted that we plotted additional data on Vieira et al (2019) in Figure 12 because that data is also worthwhile in this comparison.
We also added a summary figure in section 4.4. to explain how the data from this study compares with the previous study. The revised text and figure are follows:

Section 4.4.

[revised manuscript text omitted]

Minor comments

**8)** RC2 Considering color figure 2 by the $\delta^{18}O$ values and using different symbols to distinguish the different ice stations.

As the reviewer suggested, we made a color figure by the $\delta^{18}O$ values. Data on ice stations were presented by different symbols. The previous Figure 2 was moved to a supplementary figure. The revised figure and its caption are as follows:

[Figure]

"**Figure 2** Temperature versus salinity diagram on the surface with stations sampled in the Arctic's Laptev and East Siberian Seas. The color scale shows the $\delta^{18}O$ values in each water sample. The temperature and salinity ranges of Surface Polar Mixed Water and Atlantic Water are indicated by the area surrounded by dashed and solid lines, respectively. "

**9) RC2** The differences between total dissolved iron (TdFe) and total dissolved manganese (TdMn) should be clearly defined in the methods section. A detailed explanation is not provided until line 184. It should be clarified that total dissolved Fe includes both dissolved and particulate Fe in the water column.

As the reviewer suggested, we defined it in the method section of 2.1. The revised text is as follows:

Section 2.1.

"Seawater was collected from the side of the ship at a depth of approximately 10 m using a peristaltic pump (Geopump, Geotech Environmental Equipment, USA) and a Tygon tube. Filtered and unfiltered samples of Fe and Mn were obtained in this study to assess the labile particulate fractions of these metals. Samples of the dissolved fractions of Fe and Mn (dFe and dMn) were collected into LDPE bottles after filtration through Acropak filters connected to a Tygon tube. The samples for total dissolvable Fe and Mn (TdFe and TdMn) were collected without filtration into LDPE bottles. The pH for Fe and Mn samples was adjusted to < 1.8 by adding ultrapure grade 6 M hydrochloric acid (Tamapure AA-100, Tama Chemicals, Japan), and they were stored for a year before the analysis. The concentration differences between the unfiltered (i.e., TdFe and TdMn) and filtered samples (i.e., dFe and dMn) were therefore attributed to the acid-labile particulate fraction. "

**10) RC2** Line 257–260: The phrase "deviated towards" is confusing in this context. The surface water is not close to the meteoric $\delta^{18}O$ values, although the salinity is close. It seems you might mean "deviated from" instead. Please revise this sentence for clarity.

As the reviewer pointed out, we revised the text as "deviated from".

**11)** RC2 Definition of fmw: On line 306, clarify that fmw includes river runoff and local precipitation in the calculation of water mass fractions. (Please refer to Major Comment 3)

Following the reviewer's indication, we included the definition of $f_{mw}$ in the section on the calculation of water mass fractions.

**12)** RC2 Figure 7b and Line 319: I am curious about the r and p values for AMW and PMW separately. Please include these in the figure as you have done in Figure 7a, even if they are not statistically significant.

As the reviewer suggested, we included the r and p values in the figure as follows:

[Figure]

"**Figure 8** Plots of dFe, dMn, visible fluorescence, and UVA fluorescence in Surface Atlantic Mixed Water (AMW) and Surface Polar Mixed Water (PMW) against fractional meteoric water ($f_{mw}$) in (a)−(d) and sea ice meltwater ($f_{sim}$) in (e)−(h). The color scale shows the $N^*$ values of each water sample. Linear relationships were evaluated based on Pearson correlation coefficients ($r$). The liner fits of the visible fluorescence-$f_{mw}$ relationships in the AMW and PMW are shown by dashed and dotted lines in (c), respectively."

**13)** RC2 Figure 10b: Please report the r and p values in Figure 10b, as you did in Figure 10a, even though the correlations are not statistically significant.

As the reviewer suggested, we included the r and p values in the figure as follows:

[Figure]

"**Figure 11** Relationships of visible fluorescence with (a) dFe and (b) dMn in Surface Atlantic Mixed Water (AMW), Surface Polar Mixed Water (PMW), sea ice, and snow. Linear relationships were evaluated based on Pearson correlation coefficients ($r$), except for an outlier (▲) in (a). The linear fits of the relationships in the AMW and PMW are shown by dashed and dotted lines in (a), respectively."

---

## Author Response (AR1)

EGUSPHERE-2024-1834 | Research article

Revised to *Biogeosciences*

**Spatial distributions of iron and manganese in surface waters in the Arctic's Laptev and East Siberian seas**

Naoya Kanna, Kazutaka Tateyama, Takuji Waseda, Anna Timofeeva, Maria Papadimitraki, Laura Whitmore, Hajime Obata, Daiki Nomura, Hiroshi Ogawa, Youhei Yamashita, and Igor Polyakov

We are grateful for your evaluation and encouragement to improve our manuscript. The revised version is enclosed, and we have detailed responses to the reviewers' comments below.

Yours sincerely,

Naoya Kanna
* * *
Response to Anonymous Referee #1 comments (RC1)

Major comments

**1)** RC1 The in-depth characterization of the Arctic's Laptev and East Siberian Seas provided here is an essential contribution to understanding the biogeochemical and physical processes controlling trace metals budgets, especially given the recent accelerating changes in the region. The authors present a compelling mechanistic synthesis, but the manuscript requires refinement in its presentation. Overall, the manuscript is generally grammatically clean. However, it needs improvements in structure and readability.

We are grateful for evaluating our manuscript and encouraging us to improve the manuscript. We tried improving the manuscript's readability after receiving the reviewer's comments. This manuscript was received second round of English language editing after making revision.

First, the manuscript reads more like a report or book chapter, overly descriptive and lacking a clear hypothesis, or answering specific questions. Though a valuable contribution, it is unclear why readers should read it. For instance, focusing on contrasting the physical and biogeochemical controls of iron (Fe) and manganese (Mn) in the Arctic's Laptev and East Siberian Seas could provide a better framework for the study.

In response to the reviewer's comment, we revised the abstract to give a clearer hypothesis for this study and explain how we tackled our research questions. The revised abstract is as follows:

Abstract

[revised manuscript text omitted]

Second, the labeling of the figures is inadequate; nearly all figures lack titles, necessitating readers to refer to figure descriptions for context. Moreover, the figure descriptions do not adequately explain the terms used, and there is a notable presence of technical jargon throughout the study.

We have carefully and thoroughly revised the figures and their captions. All figures were provided titles for conciseness, and their descriptions were strived for clarity. Please check the revised all figures and their captions in the revised manuscripts.

Third, the manuscript overuses acronyms, which can hinder readability.

Following the reviewer's indications, we avoided using acronyms as much as we can throughout the text. For example, we decided not to use acronyms of Amundsen Basin, Makarov Basin, Chukchi Abyssal Plain, and East Siberian Sea in the text, which strongly hinder readability.

Finally, some sentences contain incomplete reasoning, which detracts from the overall coherence of the manuscript. Below, I highlight few examples.

We revised the text following the reviewer's comments below. Detailed responses to the reviewer's comments are as follows.

Minor comments

**2)** RC1 Line 165: Too many acronyms already, you don't need S and T for temperature and salinity.

We deleted them as the reviewer suggested.

**3)** RC1 Line 165: "The T of water above 25 m exceeded 0°C at 125−145 °E (Figs. S1 and S2), owing to the atmospheric radiative forcing" It is not clear what you mean in this sentence. What does radiative forcing have to do with surface ocean temperature? These two concepts are obviously related, but it is unclear how you are using them

We realized that this sentence had been over-interpreted because we did not measure atmospheric radiative forcing. Therefore, we removed this sentence from the text.

**4)** RC1 Line 190: Define the terms in your figures.

We defined the terms in the figure caption as the reviewer suggested. The revised figure and its caption are as follows:

[Figure]

"Figure 3 Spatial distributions of (a) silicate, (b) dMn, (c) TdMn, (d) dFe, (e) TdFe, and (f) DOC in the surface of the Arctic's Laptev and East Siberian seas. "

**5)** RC1 Line 215: "In contrast, a negative value of fsim along the continental slope suggests that sea ice formation is dominant in the region," which region?

We revised the sentence as follows:

Line 230:

"In contrast, a negative value of $f_{sim}$ in the Makarov Basin suggests that sea ice formation is dominant, which agrees with..."

**6)** RC1 Figure 4: Same issue with Figure 3, but you sometimes use full names. Be consistent.

We revised the terms in the figure following the reviewer's comment. We decided not to use 'Si', 'N+N', and 'P' because these terms were not used frequently throughout the manuscript. We also do not use the acronym of salinity (S). The revised figure and its caption are as follows:

[Figure]

"Figure 5 Vertical profiles of (a) salinity, (b) $\delta^{18}O$, (c) dFe, (d) TdFe, (e) dMn, (f) TdMn, (g) DOC, (h) visible fluorescence, (i) UVA fluorescence, (j) nitrate+nitrite, (k) phosphate, and (l) silicate in snow, sea ice, and under-ice water, respectively. "

**7) RC1 Line 255:** {The dMn concentration in the surface waters gradually increased with decreasing salinity, whereas the dFe concentration did not show a similar increasing trend (Fig. 5)} This statement is not a complete interpretation of figure 5, can you explain the exceptions

In response to the comments from the reviewer, we revised the sentence as follows:

Line 276:

"The dMn concentration in the surface water gradually increased with decreasing salinity (Fig. 6b).

The dFe concentration also showed a similar increasing trend in salinity ranges from 31 to 34 (Fig. 6a); however, the dFe concentration in less saline waters (salinity < 31) was independent of salinity variation."
* * *
Response to Anonymous Referee #2 comments (RC2)

This research investigates the concentrations and distributions of dissolved iron (dFe) and manganese (dMn) in the

surface waters of the Laptev and East Siberian Seas (LESS) during the late summer of 2021. The study concludes that the concentrations of dFe and dMn in the LESS are primarily controlled by river discharge and shelf sediment inputs rather than by sea ice melt. The manuscript contributes important data to the relatively understudied area of trace metal dynamics in the Arctic. The descriptions of sampling and analyzing methods are solid. While this manuscript leans more toward descriptive analysis, it still offers valuable insights into the region's biogeochemical processes. I have several comments on this manuscript:

We sincerely appreciate the reviewer for helping to improve our manuscript. We thoroughly revised it following the reviewer's comments.

Major comments

**1)** RC2 Revise the Abstract: The abstract lacks clarity and does not efficiently summarize the entire article. The wording is awkward in places, such as in the phrase "Nutrient-rich Pacific-sourced water exists...," which does not clearly convey that this water mass is dominant in the region. The conclusion section is clearer and does a better job of summarizing the findings. Please rewrite or revise the abstract for better clarity and conciseness.

Following the reviewer's comments, we revised the abstract for clarity and conciseness as follows:

Abstract

"The Arctic Laptev and East Siberian Seas (LESS) have high biogeochemical activity. Nutrient inputs associated with river runoff and shelf sediment-water exchange processes are vital for supporting primary production in the LESS. Relative to macronutrients, data on dissolved iron (dFe) and manganese (dMn), which are essential micronutrients for primary producers, have historically been sparse for LESS. Some dFe and dMn are reportedly carried in the Central Arctic by the Transpolar Drift, a major current that directly transports Eurasian shelf water, river water, and sea ice from the LESS continental margins. However, the supply of dFe and dMn to the surface waters of the LESS and the subsequent biogeochemical processes are not well constrained. In the summer of 2021, we investigated the following questions: *what are the sources of dFe and dMn to the surface layer* and *what factors control their concentrations and distributions on the LESS continental margins*? We demonstrated strong regional controls on dFe and dMn distributions based on distinct hydrographic regimes between the eastern side of the LESS (East Siberian Sea and Chukchi Abyssal Plain) and the western side (Makarov and Amundsen basins). Specifically, the East Siberian Sea and Chukchi Abyssal Plain were governed by Pacific-sourced water, and the Makarov and Amundsen basins were influenced by Atlantic-sourced water. Pacific-sourced water contained higher levels of dMn released from continental shelf sediments than Atlantic-sourced water. In contrast, elevated dFe signals were not observed, likely because sedimentary dFe was more rapidly removed from the water column through oxidation or scavenging than dMn. The impact of river water discharge on the dFe distributions of Pacific- and Atlantic-sourced water was significant. A positive correlation between the fraction of meteoric

water (river water and precipitation), dFe, and humic-like colored dissolved organic matter (CDOM) in these waters confirmed that dFe and CDOM are common freshwater sources. Terrigenous organic ligands likely stabilize Fe in the dissolved phase, which is not the case for Mn. Sea-ice melting and formation were not significant sources during the observation period. We conclude that the major sources controlling the dFe and dMn distributions in the LESS continental margins are river discharge and shelf sediment input."

**2)** RC2 Reorganize the structure. The method for calculating water mass fractions, described in lines 181–210, should be moved to Section 2 (Methods)

Following the reviewer's suggestion, we created a new subsection 2.5 in the Method section (lines 164–183) and moved the description for calculating water mass fractions there.

**3)** RC2 Lines 211–220 do not present the mass fraction of Atlantic water. This information should be included. And its also not shown in Figure 3.

As the reviewer indicated, we included the mass fraction of Atlantic water ($f_{Atlantic}$) in the figure. We made a new figure regarding the results of mass fraction analysis because the previous Figure 3 contained much information. The conclusion of the text has not been changed by this revision. The revised figure is as follows:

[Figure]

"**Figure 4** Spatial distributions of fractional (a) Pacific Water ($f_{Pacific}$), (b) Atlantic Water ($f_{Atlantic}$), (c) sea ice meltwater ($f_{sim}$), and (d) meteoric water ($f_{mw}$) in the surface of the Arctic's Laptev and East Siberian Seas. Abbreviations in (a) and (b): Surface Polar Mixed Water (PMW) and Surface Atlantic Mixed Water (AMW). "

**4)** RC2 Section 4.2.1: The sedimentary contribution to dFe and dMn is not adequately discussed.

In response to the reviewer's comment, we plotted dissolved fractions of the metals with $N^*$ as well. We also added a

discussion to the text regarding the sedimentary contribution to dFe and dM. The revised manuscript is as follows:

Lines 319−334:

"Our results showed that the $N^*$ value in PMW was much lower ($< -5$) than that in AMW (Fig. 7). Although Fe and Mn are considered to be released in the dissolved phase from reductive sediments over the Chukchi Shelf, these metals are gradually removed from the water column as the particulate phase (Jensen et al., 2020). The TdMn and dMn concentrations tended to be high in the low-$N^*$ PMW, suggesting a reductive sedimentary flux that released Mn from the Chukchi Shelf (Fig. 7b and d). Mn was more elevated at shallow shelf stations 86 and 89, which were the stations most influenced by shelf inputs. Given that the dMn to TdMn ratio in PMW is as high as $85.2 \pm 10.2\%$, Mn was primarily in the dissolved phase. In contrast, the TdFe and dFe concentrations in the PMW were relatively low (Fig. 7a and c) compared to those typically observed in the continental margin of the Arctic Ocean (Aguilar-Islas et al., 2013; Cid et al., 2011, 2012; Jensen et al., 2020; Klunder et al., 2012; Kondo et al., 2016; Nakayama et al., 2011; Nishimura et al., 2012). This is likely because Fe is removed much more rapidly from the Chukchi Shelf water column than Mn via oxidation and re-precipitation (Jansen et al., 2020; Millero et al., 1987; Vieira et al., 2019). Indeed, we observed a lower dFe to TdFe ratio ($43.6 \pm 23.8\%$) in PMW, such that Fe was primarily in the particulate phase. The relatively high Fe concentrations at stations Ice-1, 38, and 40 are likely attributable to riverine flux because these stations show a relatively high fractional $f_{mw}$ (see discussion below)."

[Figure]

"**Figure 7** Plots of (a) TdFe, (b) TdMn, (c) dFe, and (d) dMn in Surface Atlantic Mixed Water (AMW) and Surface Polar Mixed Water (PMW) against $N^*$ values."

**5)** RC2 Line 334–335: This section is unclear. Although sedimentary inputs to dFe and dMn were discussed in Section 4.2.1, the correlations between TdFe, TdMn, dFe, and dMn should be more clearly explained.

In response to the reviewer's comment, we plotted the correlations between TdFe, TdMn, dFe, and dMn in the supplement figure S6. We also explained the correlations between them in the Section 4.2. The revised text is as follows:

Lines 299−305.

"A significant correlation between dFe and dMn has been observed in the deeper waters (>3000 m) of the Amundsen and Makarov basins because scavenging removal is the dominant process in deep-water masses (Klunder et al., 2012). In the LESS surface water, Fe was not correlated with Mn in the unfiltered and filtered fractions (Fig. S6). Additional factors, such as external inputs, influence the distribution of dFe and dMn in surface waters, leading to the disappearance of the Fe-Mn relationship. Moreover, the enrichment of dMn compared to that of dFe (Fig. S6a) was observed in all sampled surface waters, suggesting the importance of the input fluxes of dMn or the preferential scavenging of dFe relative to dMn. Fe and Mn are redox-active metals that share common sources in surface water, such as sediments, dust deposition, and freshwater inputs. "

Supplement figure S6:

[Figure]

"Figure S6 (a) Plot of dMn in Surface Atlantic Mixed Water (AMW) and Surface Polar Mixed Water (PMW) against dFe. (b) Plot of TdMn against TdFe. The dashed line in (a) presents the relation of both metals found in the deeper waters (>3000 m) of the Amundsen and Makarov Basins, with a correlation: [dMn] = (0.15×[dFe])/0.75 (Klunder et al., 2012) "

**6)** RC2 Figure 6 and Line 297: What are the possible explanations for the outliers with high TdFe and TdMn? Are these stations heavily influenced by sedimentary inputs? Please address these points.

We highlighted the outliers in the figure. The outliers with high TdMn were obtained at two shallow southernmost stations on the shelf of the East Siberian Sea, and therefore the outliers were likely influenced by sedimentary inputs. For the TdFe, the sampled water seems to be influenced by the inputs of river runoff because these stations show relatively high fractional $f_{mw}$. We addressed the points as follows:

Lines 320−331:

"Although Fe and Mn are considered to be released in the dissolved phase from reductive sediments over the Chukchi Shelf, these metals are gradually removed from the water column as the particulate

phase (Jensen et al., 2020). The TdMn and dMn concentrations tended to be high in the low-$N^*$ PMW, suggesting a reductive sedimentary flux that released Mn from the Chukchi Shelf (Fig. 7b and d). Mn was more elevated at shallow shelf stations 86 and 89, which were the stations most influenced by shelf inputs. Given that the dMn to TdMn ratio in PMW is as high as $85.2 \pm 10.2\%$, Mn was primarily in the dissolved phase. In contrast, ...

The relatively high Fe concentrations at stations Ice-1, 38, and 40 are likely attributable to riverine flux because these stations show a relatively high fractional $f_{mw}$ (see discussion below). "

**7)** RC2 Section 4.4 and Figure 11: The comparison with other studies is valuable, but Figure 11 caption is not informative enough and needs to be modified. Please properly cite data sources in the figure captions. Use different symbols for different data sources instead of the same symbols, and highlight the data from this study using distinct symbols. Additionally, I suggest adding a summary figure by region to show how the data from this study compares with or complements the range of data reported in previous research.

When we use different symbols on the figure with a color ramp, the figure looks hard to see. Instead, we revised the figure with proper symbols and citations as follows:

[Figure]

"Figure 12 Spatial distributions in (a) dFe, (b) dMn, and (c) $N^*$ in surface water ($< 25$ m depth) in the Arctic Ocean. (d) Location of stations on dFe, dMn, and $N^*$ reported by this study and the previous studies."

It should be noted that we plotted additional data on Vieira et al (2019) in Figure 12 because that data is also worthwhile in this comparison.

We also added a summary figure in section 4.4. to explain how the data from this study compares with the previous study. The revised text and figure are follows:

Lines 456−503:

[revised manuscript text omitted]

Minor comments

**8)** RC2 Considering color figure 2 by the δ¹⁸O values and using different symbols to distinguish the different ice stations.

As the reviewer suggested, we made a color figure by the δ¹⁸O values. Data on ice stations were presented by different symbols. The previous Figure 2 was moved to a supplementary figure. The revised figure and its caption are as follows:

[Figure]

"**Figure 2** Temperature versus salinity diagram on the surface with stations sampled in the Arctic's Laptev and East Siberian seas. The color scale shows the $\delta^{18}O$ values in each water sample. The temperature and salinity ranges of Surface Polar Mixed and Atlantic waters are indicated by the area surrounded by dashed and solid lines, respectively. "

**9)** RC2 The differences between total dissolved iron (TdFe) and total dissolved manganese (TdMn) should be clearly defined in the methods section. A detailed explanation is not provided until line 184. It should be clarified that total dissolved Fe includes both dissolved and particulate Fe in the water column.

As the reviewer suggested, we defined it in the method section of 2.1. The revised text is as follows:

> Lines 88−95:
>
> "Seawater was collected from a depth of approximately 10 m from the side of the ship using a peristaltic pump (Geopump, Geotech Environmental Equipment, USA) and a Tygon tube. Filtered and unfiltered Fe and Mn samples were obtained to assess their labile particulate fractions. Samples of the dissolved fractions of Fe and Mn (dFe and dMn) were collected in LDPE bottles after filtration through AcroPak filters connected to Tygon tubes. Samples for total acid dissolvable Fe and Mn (TdFe and TdMn) were collected into LDPE bottles without filtration. The pH of the Fe and Mn samples was adjusted to < 1.8 by adding ultrapure-grade 6 M hydrochloric acid (Tamapure AA-100, Tama Chemicals, Japan), and were stored for a year before the analysis. Therefore, the concentration differences between the unfiltered (i.e., TdFe and TdMn) and filtered (i.e., dFe and dMn) samples were attributed to the acid-labile particulate fraction."

**10)** RC2 Line 257–260: The phrase "deviated towards" is confusing in this context. The surface water is not close to the meteoric $\delta^{18}O$ values, although the salinity is close. It seems you might mean "deviated from" instead. Please revise this sentence for clarity.

Lines 282–285: As the reviewer pointed out, we revised the phrase as "deviated from".

**11)** RC2 Definition of fmw: On line 306, clarify that fmw includes river runoff and local precipitation in the calculation of water mass fractions. (Please refer to Major Comment 3)

Following the reviewer's indication, we included the definition of $f_{mw}$ in the section on the calculation of water mass fractions.

**12)** RC2 Figure 7b and Line 319: I am curious about the r and p values for AMW and PMW separately. Please include these in the figure, as you have done in Figure 7a, even if they are not statistically significant.

As the reviewer suggested, we included the r and p values in the figure as follows:

[Figure]

"**Figure 8** Plots of dFe, dMn, visible fluorescence, and UVA fluorescence in Surface Atlantic Mixed Water (AMW) and Surface Polar Mixed Water (PMW) against fractional meteoric water ($f_{mw}$) in (a)−(d) and sea ice meltwater ($f_{sim}$) in (e)−(h). The color scale shows the $N^*$ values of each water sample. Linear relationships were evaluated based on Pearson correlation coefficients ($r$). The liner fits of the visible fluorescence-$f_{mw}$ relationships in the AMW and PMW are shown by dashed and dotted lines in (c), respectively."

**13)** RC2 Figure 10b: Please report the r and p values in Figure 10b, as you did in Figure 10a, even though the correlations are not statistically significant.

As the reviewer suggested, we included the r and p values in the figure as follows:

[Figure]

"**Figure 11** Relationships of visible fluorescence with (a) dFe and (b) dMn in Surface Atlantic Mixed Water (AMW), Surface Polar Mixed Water (PMW), sea ice, and snow. Linear relationships were evaluated based on Pearson correlation coefficients ($r$), except for an outlier (▲) in (a). The linear fits of the relationships in the AMW and PMW are shown by dashed and dotted lines in (a), respectively."